# Impaired vitamin D signaling reveals neutrophils as key drivers of prostate cancer dissemination

Kateryna Len-Tayon [1,2,3,4], Justine Gantzer[1,2,3,4,5], Charles Dariane[6], Olivier Fogel [7], Vanessa Friedrich [1,2,3,4], Daniela Rovito [1,2,3,4], Véronique Lindner[8], Valentine Gilbart [1,2,3,4], Darya Yanushko [1,2,3,4], Sandrine Henri[9], Daniel Metzger [1,2,3,4] & Gilles Laverny [1,2,3,4] ✉

## Abstract

Prostate cancer (PCa)-related deaths are mainly due to metastasis. The increase in de novo metastatic hormone-naive PCa (mHNPCs) highlights the urgent need for biomarkers and treatment strategies. We report in a cohort of French PCa patients that the levels of vitamin D and of prostate-specific antigen, the progression biomarker used clinically, are negatively associated. However, the impact of vitamin D receptor (VDR) signaling on prostate tumorigenesis remains unclear. Mice with PTEN inactivation in prostatic epithelial cells (PECs) at adulthood ($Pten^{(i)pe-/-}$ mice) faithfully recapitulate the human disease. We showed that inactivation of PTEN and VDR in PECs promotes tumor aggressiveness. We demonstrate that VDR loss induces oxidative stress, which in turn enhances PECs proliferation. Moreover, CXCL5 overexpression in PTEN- and VDR-deficient PECs promotes neutrophil infiltration. Importantly, our data highlight elevated circulating neutrophil levels as a biomarker of PCa dissemination and show the potency of targeting neutrophil chemotaxis to reduce liver micrometastases. Overall, this work provides major insights into how vitamin D signaling slows down tumorigenesis and opens new avenues for therapeutic and diagnostic strategies for mHNPC.

Subject Categories Cancer; Metabolism; Urogenital System

## Introduction

Prostate cancer (PCa) is the most frequent visceral neoplasm and the second leading cause of cancer-related death in men in Western society (Sung et al, 2021; Siegel et al, 2025). PCa initiates by hyperproliferation of prostatic epithelial cells (PECs), predominantly luminal cells, leading to the formation of prostatic intraepithelial neoplasia (PIN). While some PINs remain dormant for decades, others resume growth, leading to invasive adenocarcinoma (Rebello et al, 2021; Wang et al, 2009). PCa-related deaths are mainly due to metastasis, either de novo at diagnosis (synchronous metastasis) or as a recurrence years after local treatment (metachronous metastasis) (Ni et al, 2024; Gandaglia et al, 2014; Pouessel et al, 2007). The progression to metastatic disease occurs in 20% of patients after radical prostatectomy or radiation therapy. The hallmark of treatment of such metastatic hormone-naive prostate cancer (mHNPC) relies on androgen deprivation therapy (ADT) to first prevent the progression of the disease. However, this strategy eventually leads to metastatic castration-resistant PCa (mCRPC) with a poor outcome. In the last decade, the use of androgen receptor pathway inhibitors (ARPIs) in combination with ADT earlier in the natural history of the disease improves survival of both mHNPC and mCRPC patients (Cornford et al, 2024; Lowrance et al, 2023; Ploussard et al, 2024; Schaeffer et al, 2024). Importantly, following a decline in PCa screening worldwide, there is a rising incidence of synchronous metastases (De Velasco Oria De Rueda et al, 2022; Corres-Mendizabal et al, 2024; Helgstrand et al, 2018), highlighting the urgent need to gain insight into the mechanisms underlying PCa progression to identify candidate biomarkers and drug targets.

Recent preclinical research boosted by single-cell technologies has uncovered a key role of the tumor microenvironment (TME) in PCa progression (Graham et al, 2024; Yanushko et al, 2025; Pervizou et al, 2025). During prostatic tumorigenesis, the transformation of PECs leads to the activation of surrounding stromal cells, extracellular matrix (ECM) remodeling, secretion of pro-tumorigenic cytokines, and recruitment of immune cells. The PCa immune microenvironment is commonly described as "cold" due to a low infiltration of cytotoxic T cells, along with the presence of immunosuppressive neutrophils (previously known as myeloid-derived suppressor cells) that drive T cell exhaustion and contribute to immunotherapy resistance (Stultz and Fong, 2021; Calcinotto et al, 2018). Thus, even though the role of the TME in PCa remains unclear, it represents promising novel therapeutic targets for metastatic PCa.

Epidemiological studies indicate an association between low circulating vitamin D levels and PCa progression (Song et al, 2018;

[1]Institute of Genetics and Molecular and Cellular Biology (IGBMC), Illkirch-Graffenstaden 67400, France. [2]CNRS UMR 7104, Illkirch-Graffenstaden 67400, France. [3]Inserm U1258, Illkirch-Graffenstaden 67400, France. [4]University of Strasbourg, Illkirch-Graffenstaden 67400, France. [5]Department of Medical and Surgical Oncology & Hematology, Hôpitaux Universitaires de Strasbourg, Strasbourg 67000, France. [6]Department of Urology, Hôpital Européen Georges-Pompidou, AP-HP, Paris-Paris Cité University-U1151 Inserm-INEM, Necker, Paris, France. [7]Université Paris Descartes, service de rhumatologie, hôpital Cochin, AP-HP, Paris, France. [8]Département de Pathologie, Les Hôpitaux Universitaires de Strasbourg, Strasbourg 67000, France. [9]Centre d'Immunologie de Marseille-Luminy, Aix-Marseille Université, Faculté des Sciences de Luminy, Marseille, France. ✉E-mail: laverny@igbmc.fr

Murphy et al, 2014). In addition, the Vitamin D and Omega-3 trial (VITAL) highlighted that long-term vitamin D supplementation reduces PCa patient mortality (Manson et al, 2020). Moreover, the bioactive vitamin D [1α,25-dihydroxy-vitamin $D_3$ (1,25$D_3$)] has been shown to slow-down the progression of prostatic precancerous lesions in a mouse model of PCa (Banach-Petrosky et al, 2006). The effects of 1,25$D_3$ are mediated by binding to the vitamin D nuclear receptor (VDR, NR1I1) (Pike and Haussler, 1979; McDonnell et al, 1987; Rochel et al, 2000). VDR is expressed in many cell types and has anti-inflammatory and antiproliferative activities in various cancers (Bikle and Christakos, 2020; Feldman et al, 2014; Len-Tayon et al, 2024). Importantly, low VDR levels in PCa are associated with a higher patient mortality (Hendrickson et al, 2011). Previous studies using transgenic mice expressing a truncated SV40 Large T antigen protein in prostates showed that a low vitamin D diet, as well as reduced VDR signaling, promotes prostate carcinogenesis and adenocarcinoma formation (Fleet et al, 2019). In addition, VDR inactivation in metastatic PCa cell lines induces apoptosis by increasing mitochondrial activity and reactive oxygen species (ROS) production (Blajszczak and Nonn, 2019; Ricca et al, 2018). However, the consequences of prostatic epithelial VDR inactivation on tumor development and progression, as well as on PCa microenvironment, remain to be determined.

The genetically engineered $Pten^{(i)pe-/-}$ mice, in which Pten, a tumor suppressor frequently mutated in PCa, is selectively inactivated in PECs at adulthood, recapitulate the human disease progression (Ratnacaram et al, 2008). Longitudinal studies revealed that the proliferation of PTEN-deficient PECs is enhanced between 1 and 3 months after gene inactivation (AGI) to form PINs, followed by a progressive growth arrest of PECs with characteristics of cellular senescence, as well as an immune cell infiltration (Parisotto et al, 2018; Pervizou et al, 2025). However, while most PINs do not progress, some PTEN-deficient PECs actively proliferate to form invasive adenocarcinoma. In addition, we showed that Luminal A (Lin⁻/CD49f⁺/Sca1⁻) to Luminal C (Lin⁻/CD49f⁺/Sca1⁺) cell plasticity underlies tumor evolution (Abu El Maaty et al, 2022).

To gain an in-depth understanding of the role of vitamin D signaling in prostate tumorigenesis, we analyzed the association of vitamin D levels with various clinical parameters in a cohort of French patients with de novo PCa, as well as the impact of VDR loss in PECs of $Pten^{(i)pe-/-}$ mice.

# Results

## Correlation of vitamin D and prostatic-specific antigen levels in a French PCa patient cohort

The identification of predictive markers contributing to PCa progression is an unmet need. In a cohort of 77 newly diagnosed PCa patients from the Paris University Hospital, treated with hormonotherapy for less than 2 months and no vitamin D or calcium supplementation (Table 1), we analyzed the correlation of various clinical parameters with serum prostatic specific antigen (PSA) levels, the biomarker of prostate cancer diagnosis (Rebello et al, 2021). Whereas the body mass index (BMI), age, or circulating neutrophil-to-lymphocyte ratio did not correlate with PSA levels (Fig. 1A), those of serum alkaline phosphatase (ALP) were

**Table 1. General characteristics of the analyzed PCa patients from the HormOS cohort.**

| Characteristics of the cohort | Mean ± SD |
|---|---|
| Total cases | $n = 77$ |
| Age | 74.5 +/− 8.5 |
| Serum PSA, ng/ml | 28.7 +/− 49.5 |
| Serum 25(OH)D, ng/ml | 13.5 +/− 8.7 |
| BMI | 27.4 +/− 4.3 |
| Leukocytes (c/μl) | 6109.1 +/− 1915.1 |
| **Clinical TNM tumor stratification** | No. of cases (%) |
| T1-T2 (N0, M0) | 29 (37.7%) |
| T3-T4 (N0, M0) | 9 (11.7%) |
| N1 (M0) | 21 (27.3%) |
| M (M1a,b,c) | 17 (22.1%) |
| ND | 1 (1.3%) |
| **ISUP (Gleason score)** | No. of cases (%) |
| ISUP 1 | 2 (2.6%) |
| ISUP 2 | 24 (11.7%) |
| ISUP 3 | 39 (31.2%) |
| ISUP 4 | 24 (20.8%) |
| ISUP 5 | 48 (32.5%) |
| ND | 1 (1.3%) |
| **Serum PSA levels (ng/ml)** | No. of cases (%) |
| <10.0 | 45 (63.8%) |
| 10.1–20.0 | 11 (14.3%) |
| >10.0 | 17 (22.1%) |
| ND | 5 (6.5%) |
| **Serum 25(OH)D levels (ng/ml)** | No. of cases (%) |
| <21 | 48 (62.3%) |
| 21–30 | 20 (26.0%) |
| >30 | 9 (11.7%) |

positively associated ($p = 0.008$, $r_P = 0.31$) (Fig. 1A,B), as recently reported (Liaqat et al, 2024). Moreover, while serum levels of the metabolite clinically used to monitor vitamin D levels [25-hydroxy-vitamin $D_3$ (25(OH)$D_3$)] were similar in patients bearing tumors with an International Society of Urological Pathology (ISUP) score of 1–2 and of 3-5 (Appendix Fig. S1), they were negatively associated with those of PSA ($p = 0.006$, $r_P = -0.32$) (Fig. 1A,C). In contrast to ALP produced by osteoblasts (Liaqat et al, 2024), PSA is secreted by PECs. In addition, VDR levels in protein extracts from prostates of $Pten^{(i)pe-/-}$ mice 1 month AGI were 5-fold higher than in those from control mice (Fig. EV1A,B). Taken together, these data suggest that vitamin D signaling in PECs impacts PCa progression.

## Impact of VDR-loss on PTEN-deficient PECs

To challenge this hypothesis, mice in which PTEN and VDR are selectively inactivated in PECs were generated (Fig. EV1C). In contrast to

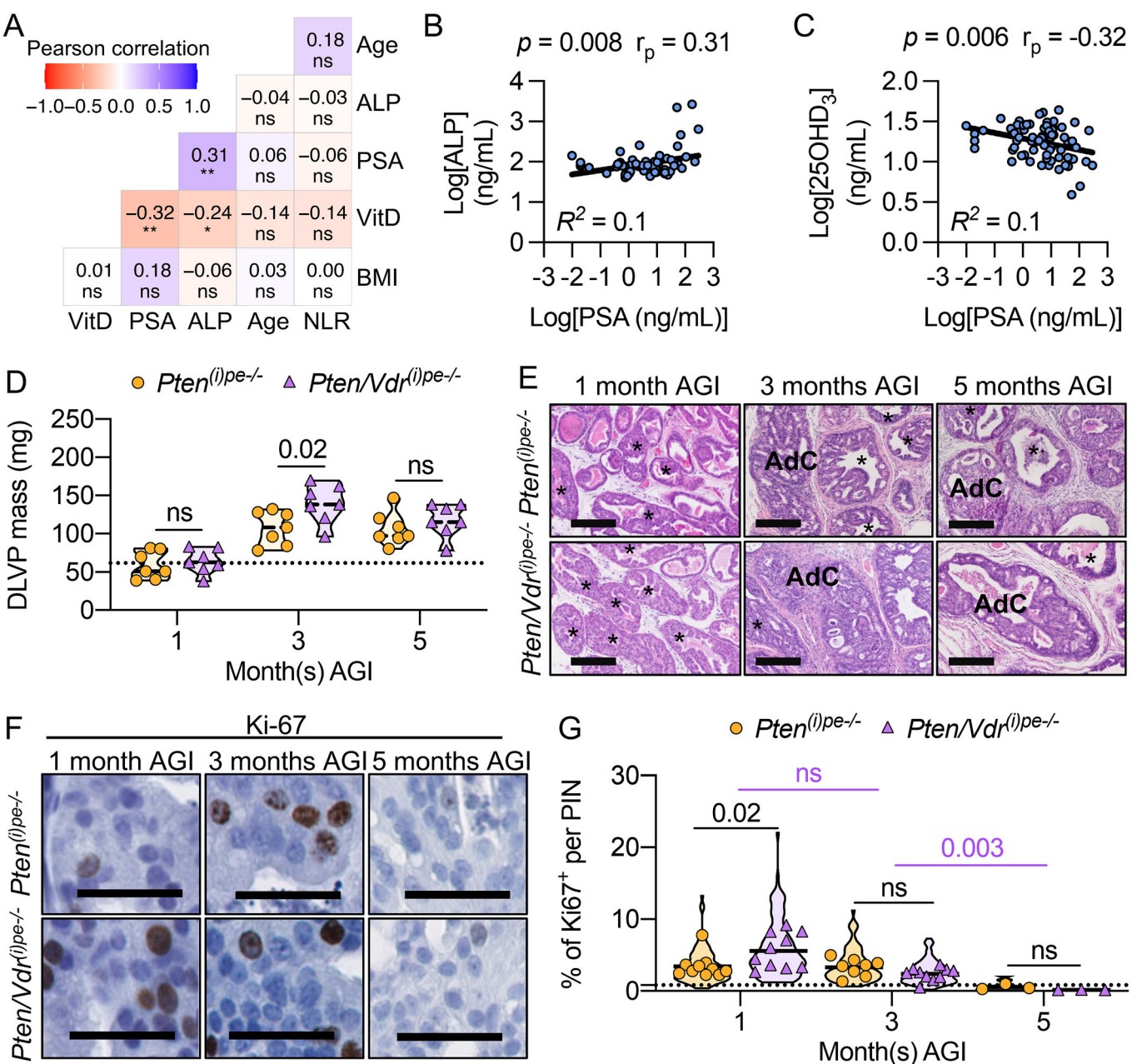

**Figure 1. Identification of predictive markers and impact of VDR-loss on PCa progression.**

(A) Representative Pearson' multivariate regression analysis of body mass index (BMI), blood 25(OH)D$_3$ (VitD), prostate specific antigen (PSA) and alkaline phosphatase (ALP) levels, age and circulating neutrophil-to-lymphocyte ratio (NLR) in newly diagnosed untreated PCa patients. $n = 77$. *$p < 0.05$, ** $p < 0.01$. Linear regression of PSA levels with ALP (B) and 25(OH)D$_3$ (C) levels. Coefficient of determination ($R^2$), $p$ values and Pearson correlation coefficient ($r_p$) are indicated. (D) DLVP mass of $Pten^{(i)pe-/-}$ and $Pten/Vdr^{(i)pe-/-}$ mice, 1, 3 and 5 month(s) AGI. $n = 7$ mice/group. The $p$ values determined by two-way ANOVA with Tukey's post-hoc are indicated. ns, $p \geq 0.05$. The dotted line represents the mean of DLVP mass from age-matched $Pten/Vdr^{L2/L2}$ mice. (E) Representative HE staining of DLVP sections from $Pten^{(i)pe-/-}$ and $Pten/Vdr^{(i)pe-/-}$ mice 1, 3 and 5 months AGI. Scale bar = 250 μm. $n = 7$ mice/group. *, PIN; AdC, adenocarcinoma. Representative immunostaining (F) and quantification (G) of Ki-67 on DLVP sections from $Pten^{(i)pe-/-}$ and $Pten/Vdr^{(i)pe-/-}$ mice 1, 3 and 5 months AGI. Scale bar = 50 μm; $n = 10$ mice/group (1 and 3 months AGI) and $n = 3$ (5 months AGI). 10 PINs/section. The $p$ values determined by two-way ANOVA with Tukey's post-hoc are indicated. ns, $p \geq 0.05$. The dotted line represents the mean of Ki-67 positive cells in the DLVPs from age-matched $Pten/Vdr^{L2/L2}$ mice. Source data are available online for this figure.

tamoxifen-treated $Pten/Vdr^{L2/L2}$ mice, the recombined PTEN and VDR alleles (L-) were detected in the prostate of PSA-CreER$^{T2}$/$Pten/Vdr^{L2/L2}$ mice 1 month after tamoxifen administration (Fig. EV1D). Moreover, VDR levels were similar in tamoxifen-treated $Pten/Vdr^{L2/L2}$ and tamoxifen-treated PSA-CreER$^{T2}$/$Pten/Vdr^{L2/L2}$ mice (Fig. EV1A,B),

suggesting that the enhanced VDR levels observed in $Pten^{(i)pe-/-}$ mice originate from PECs. Therefore, tamoxifen-treated PSA-CreER$^{T2}$/$Pten/Vdr^{L2/L2}$ mice were termed $Pten/Vdr^{(i)pe-/-}$, and tamoxifen-treated $Pten/Vdr^{L2/L2}$ littermates were used as controls. Histological analysis revealed the presence of PIN-positive for PTEN-loss-induced AKT

phosphorylation (p-AKT Ser$^{473}$) in the dorsolateral and ventral prostates (DLVP) of $Pten^{(i)pe-/-}$ and $Pten/Vdr^{(i)pe-/-}$ mice, whereas only few p-AKT+ epithelial cells were observed in the anterior prostate (AP) (Fig. EV1E). These results demonstrate that $Pten$ is efficiently inactivated in DLVPs of $Pten^{(i)pe-/-}$ and $Pten/Vdr^{(i)pe-/-}$ mice, whereas $Vdr$ is selectively inactivated in the latter.

To gain insight into the impact of VDR-loss on PIN formation, DLVP of $Pten^{(i)pe-/-}$ and $Pten/Vdr^{(i)pe-/-}$ mice were analyzed at various timepoints. At 1 month AGI, both had a prostate mass of approximatively 50 mg as seen in the controls, but histological analysis showed that 50% of the glands in $Pten^{(i)pe-/-}$ and $Pten/Vdr^{(i)pe-/-}$ mice contain PIN, with a similar average gland area (Figs. 1D,E and EV2A,B). Moreover, flow cytometry analysis of Lin-negative cells in DLVPs revealed a similar proportion of Luminal A and Luminal C cells in $Pten^{(i)pe-/-}$ and $Pten/Vdr^{(i)pe-/-}$ mice (Appendix Fig. S2A–C). However, the percentage of PECs positive for the proliferation marker Ki-67 (Ki-67 +) in DLVPs was higher in $Pten/Vdr^{(i)pe-/-}$ than in $Pten^{(i)pe-/-}$ mice (Fig. 1F,G). To identify the mechanism underlying the enhanced proliferation in PTEN-deficient PECs following VDR loss, RNAseq was performed on FACS-isolated luminal cells from the DLVP (Appendix Fig. S2D). The analysis revealed 2449 differentially expressed transcripts (979 up- and 1470 down-regulated) between $Pten/Vdr^{(i)pe-/-}$ and $Pten^{(i)pe-/-}$ mice (Fig. 2A; Dataset EV1). Consistent with the increased proliferation in PECs of $Pten/Vdr^{(i)pe-/-}$ mice, the transcript levels of genes associated with cell growth ($Pik3ca, Akt1$) were higher in luminal cells of $Pten/Vdr^{(i)pe-/-}$ mice, whereas those related to cell cycle arrest ($Cdkn1a$) were lower (Fig. 2B). Furthermore, pathway analysis using the Kyoto Encyclopedia of Genes and Genomes (KEGG) database revealed that the down-regulated genes in VDR-deficient luminal cells are associated to focal adhesion and extracellular matrix (ECM)-receptor interaction (Thbs4/Itga11/Itgb3/Lamc1) (Dataset EV2), whereas those up-regulated are mainly involved in oxidative phosphorylation (OXPHOS), including mitochondrial complexes I ($Ndufb8$), II ($Sdhb, Sdhd$), III ($Uqcrq$), ATP synthase ($Atp6v1b1$), reactive oxygen species (ROS) ($Mgst2, Sos2$) and glutathione metabolism ($Gsta4/Gstm6$) (Fig. 2B,C; Dataset EV2), indicating that VDR loss enhances oxidative stress (OS) in PTEN-deficient PECs. Indeed, immunostaining showed that the percentage of PECs positive for 8-hydroxy-2'-deoxyguanosine (8-OHdG), a key marker of OS-induced DNA adducts (Murphy et al, 2022), was higher in the DLVPs of $Pten/Vdr^{(i)pe-/-}$ mice than in those of $Pten^{(i)pe-/-}$, 1 month AGI (Fig. 2D,E). The number of GC to TA transversion compared to the Mus musculus reference genome in the luminal cells from the DLVP of $Pten^{(i)pe-/-}$ and $Pten/Vdr^{(i)pe-/-}$ mice 1 month AGI was similar (Appendix Fig. S3A), and the terminal deoxynucleotidyl transferase dUTP nick end labeling (TUNEL) assay did not reveal DNA fragmentation in PECs of $Pten/Vdr^{(i)pe-/-}$ mice within the first months AGI (Fig. EV2D), indicating that OS does not promote 8-OHdG-induced transversion nor apoptosis. To further evaluate the impact of OS on PIN, $Pten/Vdr^{(i)pe-/-}$ mice 2 days AGI were treated with the ROS scavenger N-acetylcysteine (NAC) (Meyer et al, 2017) for 4 weeks (Fig. 2F). Importantly, the percentage of 8-OHdG+ cells was reduced by 2-fold in NAC-treated mice, and that of Ki-67+ cells returned to basal levels (Fig. 2G–J).

To confirm the role of VDR-loss on OS-induced PEC proliferation, VDR and/or PTEN were silenced in non-malignant human prostatic epithelial (RWPE-1) cells using siRNA-based approaches (Fig. 2K; Appendix Fig. S3B). PTEN-silencing had almost no effect on mitochondrial respiration, but enhanced the proliferation of RWPE-1 cells (Fig. 2L–N; Appendix Fig. S3C). Importantly, VDR levels were increased after PTEN-loss (Fig. 2K), confirming that VDR overexpression is a regulatory feedback following PTEN inactivation in PECs. In contrast to PTEN-silencing alone, and as suggested by the enhanced transcript levels of genes encoding proteins of the mitochondrial respiratory chain (Fig. 2B), combined PTEN- and VDR-silencing promoted both basal and maximal respiration (Fig. 2L; Appendix Fig. S3C), indicating that mitochondria contribute to VDR-loss induced OS. In agreement with the results obtained in $Pten^{(i)pe-/-}$ mice, concomitant VDR-silencing further increased the proliferation of PTEN-silenced cells (Fig. 2M,N). Moreover, the proliferation of PTEN- and VDR-silenced cells was similar to that of control-silenced cells in the presence of NAC (Fig. 2M,N). Taken together, these results demonstrate that VDR levels are increased in PECs after PTEN inactivation, and that oxidative stress is enhanced when VDR is absent, which, in turn, promotes the proliferation of PTEN-deficient PECs.

## VDR-loss in PTEN-deficient PECs impacts the tumor microenvironment

The DLVP mass increased in $Pten^{(i)pe-/-}$ and $Pten/Vdr^{(i)pe-/-}$ mice from 1 to 3 months AGI, but was 40% higher in the latter (Fig. 1D), in line with the higher proliferation rate observed 1 month AGI. Histological scoring showed a similar gland area and architecture—10% of neoplastic-free glands, 75% PIN-containing glands, and about 15% adenocarcinoma (AdC) revealed by discontinuous smooth muscle layers around some prostate ducts (Pan et al, 2023)—in DLVP of $Pten^{(i)pe-/-}$ and $Pten/Vdr^{(i)pe-/-}$ mice (Figs. 1E and EV2A–C). In addition, 90% of luminal cells were of the C subtype in DLVPs from $Pten^{(i)pe-/-}$ and $Pten/Vdr^{(i)pe-/-}$ mice (Appendix Fig. S2C,E,F). Most of them were positive for SA-β-gal activity, whereas none were stained 1 month AGI (Fig. EV2E). Moreover, the levels of Ki-67+ PECs were similar at 3 months AGI, but 10-fold reduced in the mutant lines 5 months AGI (Fig. 1F,G), and the DLVPs mass and histology were similar 3 and 5 months AGI in $Pten^{(i)pe-/-}$ and $Pten/Vdr^{(i)pe-/-}$ mice (Figs. 1D,E and EV2A,B). Thus, while VDR-loss increased the prostate mass of $Pten^{(i)pe-/-}$ mice 3 months AGI, it does not affect the proliferation or SA-ßGal activity in PTEN-null PECs.

To determine the consequences of epithelial VDR-loss on PECs and tumor microenvironment heterogeneity 3 months AGI, single-cell RNA sequencing (scRNA-seq) on dissociated prostates was performed. Unbiased analysis of 5798 and 6752 cells from one $Pten^{(i)pe-/-}$ and one $Pten/Vdr^{(i)pe-/-}$ mouse, respectively, identified twenty-three cell clusters, which were further regrouped in nine clusters according to their lineage markers (Fig. 3A,B; Dataset EV3). Briefly, clusters 18 and 19 expressing the markers of the seminal vesicle epithelium ($Svs5$ and $Pate4$) were considered as contaminants and removed from further analysis (Dataset EV3). The other clusters were annotated using previously described markers (Abu El Maaty et al, 2021): neutrophil ($S100a8$, clusters 0 and 7), T cells ($Cd3e$, clusters 3, 8), macrophage ($Adgre1$, clusters 4, 11, 12, 13), B-cells ($Cd79a$, cluster 21), stromal cells ($Col1a1$, clusters 1, 2, 6, 10, 14, 15, 22), basal cells ($Krt5$, cluster 17), luminal A cells ($Krt8$, clusters 5, 9), luminal C cells ($Krt8$ and $Tacstd2$, cluster 20) and endothelial cells ($Pecam1$, cluster 16) (Fig. 3B). All clusters were identified in prostates from both mice (Appendix Fig. S4 and Dataset EV3). To determine the consequences of VDR

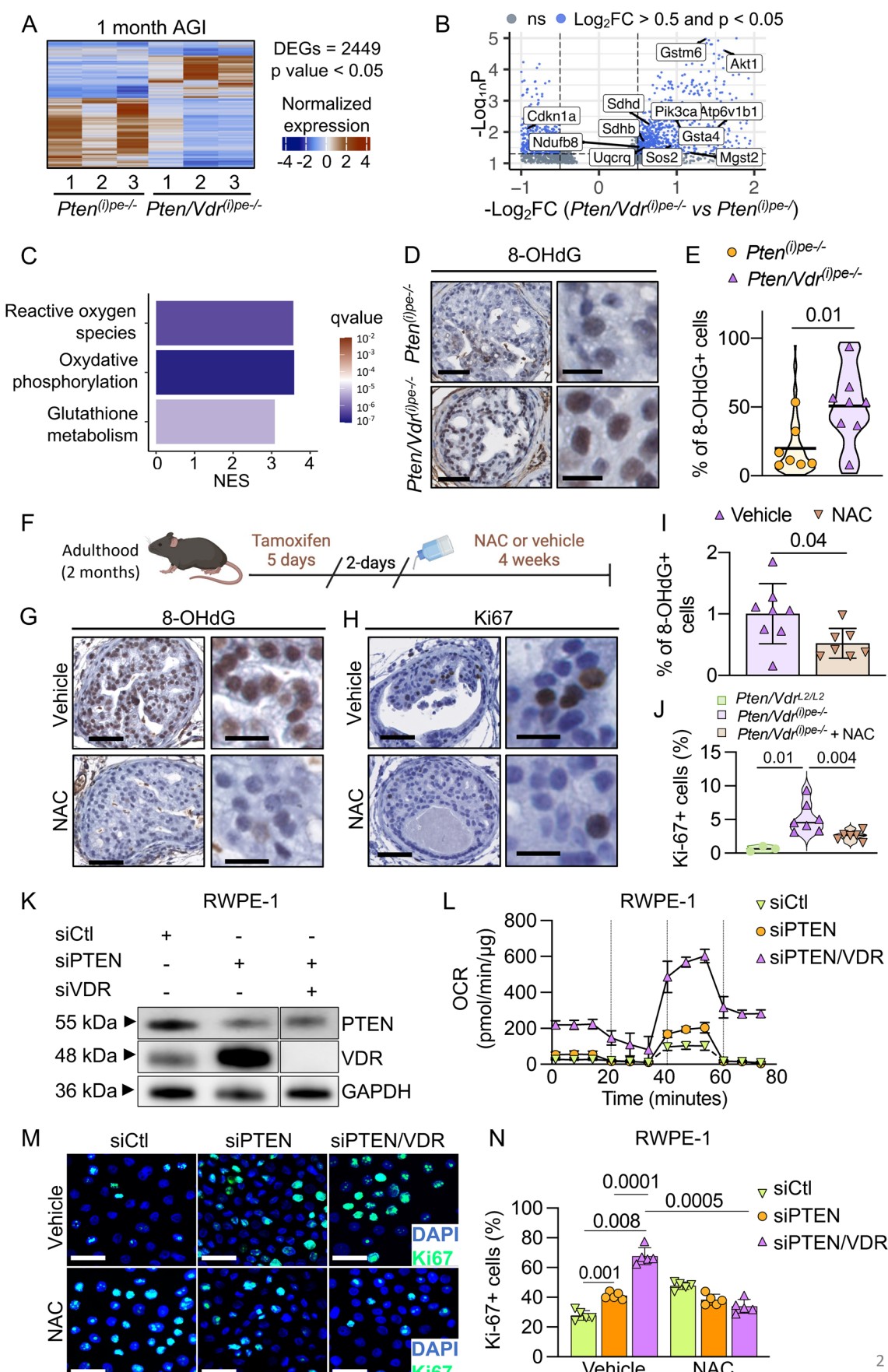

◀ **Figure 2. Oxidative stress in PTEN- and VDR-null PECs.**

Heatmap (**A**) and volcano plot (**B**) of the differentially expressed genes (DEGs) between luminal cells in DLVPs of $Pten^{(i)pe-/-}$ and $Pten/Vdr^{(i)pe-/-}$ mice, 1 month AGI determined by a Wald test corrected for multiple testing using the Benjamini and Hochberg method. ($n = 3$ mice per genotype), ns (gray dot), $p \geq 0.05$. (**C**) KEGG database annotation of the DEGs. NES: normalized enriched score. (**D**) Representative immunostaining of 8-OHdG on DLVP sections from $Pten^{(i)pe-/-}$ ($n = 7$) and $Pten/Vdr^{(i)pe-/-}$ ($n = 8$) mice 1 month AGI. Scale bar = 50 μm (left) and 20 μm (right). (**E**) Percentage of 8-OHdG positive cells per PIN (10 PINs/mouse) in DLVP sections from $Pten^{(i)pe-/-}$ ($n = 7$) and $Pten/Vdr^{(i)pe-/-}$ ($n = 8$) mice 1 month AGI. $p$ value determined by Student's $t$-test. (**F**) Schematic representation of NAC administration to $Pten/Vdr^{(i)pe-/-}$ mice. Representative immunostaining of 8-OHdG on DLVP sections from $Pten^{(i)pe-/-}$ ($n = 7$) and $Pten/Vdr^{(i)pe-/-}$ ($n = 8$) mice. Scale bar = 50 μm (left) and 20 μm (right). Percentage of 8-OHdG (**I**) and Ki-67 (**J**) positive cells per PIN (10 PINs/mouse) in DLVP sections from $Pten/Vdr^{L2/L2}$, and from $Pten^{(i)pe-/-}$ and $Pten/Vdr^{(i)pe-/-}$ mice 1 month AGI. $N = 6$–8 mice per group. $p$ value determined by Student's $t$-test. (**K**) Representative immunoblot analysis of PTEN and VDR levels in RWPE-1 cells transfected with non-targeting control (siCtrl), PTEN (siPTEN), and/or VDR (siVDR) siRNAs for 48 h. GAPDH was used as a loading control. (**L**) Representative oxygen consumption rate (OCR) in RWPE-1 cells transfected as indicated. Data are represented as mean +/− standard deviation. $n = 8$ independent replicates per condition. Representative Ki-67 (green) immunostaining (**M**) and percentage of positive cells (**N**) in RWPE-1 cells transfected as indicated and treated with either vehicle (upper panel) or 1 mM NAC (lower panel) for 48 h. Data are represented as mean + standard deviation. DAPI (blue) stains the nuclei. Scale bar, 100 μm; $n = 5$ independent replicates per condition. $p$ value determined by two-way ANOVA with Tukey's post-hoc are indicated. Source data are available online for this figure.

inactivation on Luminal C cells, we analyzed the differentially expressed genes (DEGs) (393 up- and 738 down-regulated) in this cluster (Dataset EV4). Gene annotation using the KEGG database showed that the up-regulated DEGs in VDR-deficient luminal cells are related to metabolic pathways or OXPHOS (*Gstm7, Ndufb8, Gsta3, Uqcrq, Mgat2, Gpx3*) and chemokine signaling (*Ccl2, Ccl3, Cxcl16, Nfkbia*) (Fig. 3C; Dataset EV5). scRNA-seq is a potent technology to gain insights into changes into the cell signature, but the sequencing depth and the number of mice used might limit the identification of downstream regulatory pathways. To overcome such a limitation, gain statistical power, and get an in-depth characterization of Luminal C cells, RNA-seq analysis of FACS-isolated luminal cells from three dissociated DLVPs from $Pten^{(i)pe-/-}$ and three from $Pten/Vdr^{(i)pe-/-}$ mice 3 months AGI was performed (Appendix Fig. S2G). This analysis revealed 1287 (557 up- and 730 down-regulated) DEGs in luminal DLVP cells of $Pten/Vdr^{(i)pe-/-}$ compared to $Pten^{(i)pe-/-}$ mice (Fig. 3D; Dataset EV6), with 120 genes shared with the gene set identified in Luminal C cluster by scRNA-seq (Fig. 3E, Table EV1). The down-regulated DEGs in VDR-deficient luminal cells identified by RNA-seq were not associated to specific pathways of the KEGG database (Table EV2), while those up-regulated were associated with metabolic pathways, such as cellular response to OS and glutathione metabolism (*Mgst2, Gstm4, Gsta2*), as well as DNA adducts (*Ugt2b34, Ugt1a8, Ugt1a6b*) (Fig. 3F,G; Table EV2). Altogether, these data indicate that VDR-loss has no impact on the heterogeneity of tumor cells or their microenvironment, but it contributes to sustain OS in Luminal C cells.

Importantly, *Cxcl5*, which encodes the main chemokine involved in neutrophil recruitment (Duits and De Visser, 2021), was one of the top-ranked upregulated DEGs in luminal DLVP cells of $Pten/Vdr^{(i)pe-/-}$ compared to $Pten^{(i)pe-/-}$ mice (Fig. 3G). In line with this observation, CXCL5 levels were 2-fold higher in total protein DLVP extracts of $Pten/Vdr^{(i)pe-/-}$ mice than in those of $Pten^{(i)pe-/-}$, 3 months AGI (Fig. 3H), suggesting an impact of vitamin D signaling in PECs on neutrophil chemotaxis. Intriguingly, and in line with this result, the analysis of the French PCa patient cohort suggested a negative association between circulating neutrophils and serum 25(OH)D$_3$ levels ($p = 0.05$, $r_p = -0.23$) (Fig. 3I). Moreover, while flow cytometry analysis and immunostaining of lymphocyte antigen 6 family member G (Ly-6G), the main neutrophil surface marker, revealed a marked increase of neutrophil infiltration between 1- and 3-months AGI in both mutant lines, that was 2-fold higher in DLVPs of $Pten/Vdr^{(i)pe-/-}$ mice than in those of $Pten^{(i)pe-/-}$, 3 month AGI (Figs. 3J–L and EV3A). Importantly, KEGG pathway

analysis of the DEG in the T cell cluster obtained by scRNA-seq analysis (Fig. 3A,B) showed that genes related to apoptosis, programmed death-1 (PD-1) checkpoint pathway and Th-17 differentiation (*Cd3g, Pdcd1, Fos*) are upregulated in prostates from $Pten/Vdr^{(i)pe-/-}$ mice (Fig. 3M; Dataset EV5). The PD-1 protein encoded by *Pdcd1*, the receptor for PD-L1 encoded by *Cd274*, regulates the primary immune response, and is implicated in T-cell exhaustion (Adrover et al, 2023). Flow cytometry analysis showed that the frequency of cytotoxic (CD8 +) T cells expressing PD-1 was 2-fold higher in DLVPs from $Pten/Vdr^{(i)pe-/-}$ mice than in those from $Pten^{(i)pe-/-}$ and controls (Figs. 3N and EV3B). Moreover, scRNA-seq data revealed that *Cxcl5* transcripts are selectively expressed in luminal cells, whereas those of *Cd274*, are mainly detected in the neutrophil clusters (Fig. 3O). Thus, these results showed that VDR-loss increases the expression of CXCL5 by PTEN-deficient PECs resulting in enhanced neutrophil infiltration and T cell exhaustion.

## PCa progression and dissemination in $Pten/Vdr^{(i)pe-}$ mice

As an enhanced immunosuppressive environment promotes cancer aggressiveness (Bancaro et al, 2023; Lopez-Bujanda et al, 2021; Yang et al, 2021), we characterized the tumors at later stages. Histological analysis did not reveal major tumor progression between 3 to 9 months AGI in DLVP of $Pten^{(i)pe-/-}$ and $Pten/Vdr^{(i)pe-/-}$ mice (Fig. EV2A–C). Note that PINs were formed in AP 3 months AGI and evolved into AdC around 5 months AGI in both $Pten^{(i)pe-/-}$ and $Pten/Vdr^{(i)pe-/-}$ mice (Fig. EV2F). Prostate mass (AP and DLVP) of both mouse lines was similar from 5 to 12 months AGI. Note that out of twenty mice per genotype, nine $Pten/Vdr^{(i)pe-/-}$ mice, but only one $Pten^{(i)pe-/-}$ mouse, had a prostate mass above 500 mg 12 months AGI (Fig. 4A). Histological scoring at this stage showed that about 60% of the glands contained AdC in DLVP of $Pten/Vdr^{(i)pe-/-}$ mice, but only 28% in those of $Pten^{(i)pe-/-}$ (Fig. EV2A). Note that AP cystic carcinomas were observed with a 2-fold higher prevalence in VDR-deficient tumors (Fig. 4B,C), whereas 5 and 10% of $Pten/Vdr^{(i)pe-/-}$ mice 12 months AGI exhibited squamous carcinoma and sarcomatoid tumors in the AP, respectively (Fig. 4B,C). Thus, following VDR inactivation, the frequency of these rare histologic PCa variants, commonly associated with rapid growth and poor clinical outcomes, is increased. Tumor aggressivity is often associated with metastatic spread. We determined the presence of tumor cells in livers and showed that the prevalence of $Pten/Vdr^{(i)pe-/-}$ mice with pan-cytokeratin positive (PanCK +) and CK19 negative

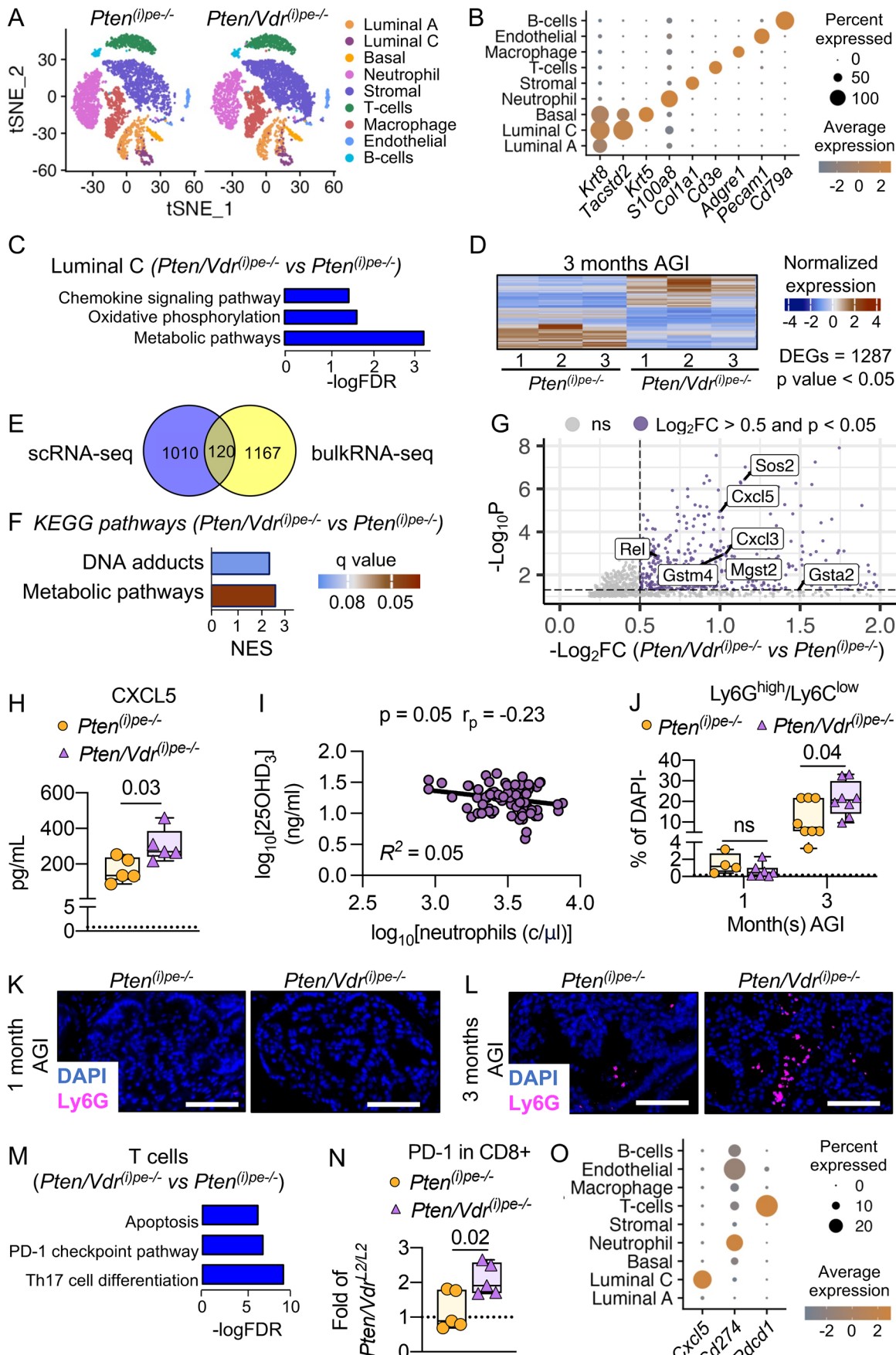

**Figure 3. VDR-loss in PTEN-deficient PECs impacts the tumor microenvironment.**

(A) t-distributed Stochastic Neighbor Embedding (t-SNE) plot of cells from dissociated prostates of $Pten^{(i)pe-/-}$ and $Pten/Vdr^{(i)pe-/-}$ mice, 3 months AGI ($n=1$ prostate per genotype). (B) Dot plot representing the expression of cell lineage markers per cluster. (C) KEGG database annotation of the DEGs in the luminal C cluster. FDR: false discovery rate. (D) Heatmap representing the DEGs between FACS-isolated luminal cells of $Pten^{(i)pe-/-}$ and $Pten/Vdr^{(i)pe-/-}$ DLVPs, 3 months AGI, determined by a Wald test corrected for multiple testing using the Benjamini and Hochberg method ($n=3$ mice per condition). (E) Venn diagram of DEGs identified by scRNA-seq (Luminal C cluster) and by bulk RNA-seq (FACS-isolated luminal cells). (F) KEGG database annotation of the DEGs obtained by bulk RNAseq analysis. NES: normalized enriched score. (G) Volcano plot of DEGs between luminal cells in DLVPs of $Pten^{(i)pe-/-}$ and $Pten/Vdr^{(i)pe-/-}$ mice, 3 months AGI, determined by a Wald test corrected for multiple testing using the Benjamini and Hochberg method ($n=3$ mice per condition). Gray dots ns, $p \geq 0.05$. (H) CXCL5 levels determined in protein extracts from DLVPs of $Pten^{(i)pe-/-}$ and $Pten/Vdr^{(i)pe-/-}$ mice 3 months AGI. The dotted line represents the mean of CXCL5 levels in protein extracts from DLVPs of age-matched $Pten/Vdr^{L2L2}$ mice. The boxes extend from the 25th to 75th percentiles, the lines represent the median, and the whiskers go down to the smallest up to the largest value. $n=5$ mice/group. The $p$ value determined by Student's $t$-test is indicated. (I) Scatter plots of circulating levels of 25(OH)D$_3$ and neutrophils in the serum of PCa patients ($n=77$). Estimated linear regression line (black), coefficient of determination ($R^2$), $p$ values and Pearson correlation coefficient ($r_p$) are indicated. (J) Flow cytometry analysis of alive (DAPI negative) Ly-6G$^{high}$/Ly-6C$^{low}$ neutrophils quantified in DLVPs 1 and 3 months AGI. $n=8$ $Pten/Vdr^{L2/L2}$, $Pten^{(i)pe-/-}$ and $Pten/Vdr^{(i)pe-/-}$ mice 3 months AGI, 4 $Pten^{(i)pe-/-}$ mice 1 month AGI and 6 $Pten/Vdr^{(i)pe-/-}$ mice 1 month AGI. The boxes extend from the 25th to 75th percentiles, the lines represent the median, and the whiskers go down to the smallest up to the largest value. The $p$ value determined by two-way ANOVA with Tukey's post-hoc are indicated. ns, $p \geq 0.05$. The dotted line represents the mean of Ly-6G$^{high}$/Ly-6C$^{low}$ neutrophils quantified in DLVPs of age-matched $Pten/Vdr^{L2L2}$ mice. Representative immunostaining of Ly-6G (magenta) on DLVP sections from $Pten^{(i)pe-/-}$ and $Pten/Vdr^{(i)pe-/-}$ mice 1 (K) and 3 (L) months AGI. Scale bar $=100$ μm. $n=3$ mice. DAPI (blue) stains the nuclei. (M) KEGG database annotation of the DEGs in the T cell cluster. FDR: false discovery rate. (N) Flow cytometry analysis of alive (DAPI-) CD8 + /PD1 + T cells quantified in DLVPs of $Pten^{(i)pe-/-}$ and $Pten/Vdr^{(i)pe-/-}$ mice, 3 months AGI represented as fold of $Pten/Vdr^{L2/L2-}$ mice. The boxes extend from the 25th to 75th percentiles, the lines represent the median, and the whiskers go down to the smallest up to the largest value. $n=5$ mice/group. The dotted line represents the mean of PD-1 +/CD8 + T cells quantified in DLVPs of age-matched $Pten/Vdr^{L2L2}$ mice. The $p$ value determined by Student's $t$-test is indicated. (O) Dot plot representing the expression of $Cxcl5$, $Cd274$ and $Pdcd1$ determined by sc-RNA seq. Source data are available online for this figure.

(a marker of cholangiocytes) cell dissemination reaches 80%, compared to 38% of $Pten^{(i)pe-/-}$ mice 12 months AGI (Figs. 4D–F and EV4A). Importantly, at 9 months AGI, a stage at which no sarcomatoid tumor is detected, cell infiltrates, composed of immune cells (CD45 +), endothelial cells (CD31 +) and PanCK+ epithelial cells, were observed in the liver of 54% of $Pten/Vdr^{(i)pe-/-}$ mice, but not in those of $Pten^{(i)pe-/-}$ mice (Fig. 4D). In agreement with a previous study (Hawley et al, 2023), immunostaining on liver sections from $Pten/Vdr^{(i)pe-/-}$ mice 9 months AGI demonstrated that these infiltrates are mainly composed of F4/80+ macrophages (24%), predominantly located at the periphery, Ly-6G+ neutrophils (2.8%), and CD3 + T cells (37%), which are evenly distributed within the infiltrates (Fig. 4G,H). Thus, VDR-loss enhances the presence of liver micrometastases characterized by a high immune infiltrate containing neutrophils.

## Neutrophils involvement in PECs dissemination to the liver

As neutrophils are known to favor tumor spreading, we determined their circulating and prostate-infiltrating levels at 9 months AGI. Immunophenotyping of DLVP and blood showed that the proportion of infiltrated neutrophils (Fig. 5A) and of exhausted CD8 + T cells (Fig. 5B) in prostates was similar in $Pten^{(i)pe-/-}$ and $Pten/Vdr^{(i)pe-/-}$ mice, whereas the number of circulating neutrophil was 1.5-fold higher in $Pten/Vdr^{(i)pe-/-}$ than in $Pten^{(i)pe-/-}$ mice (Figs. 5C and EV4B). Further analysis did not reveal an association between circulating and prostate-infiltrated neutrophils in $Pten/Vdr^{(i)pe-/-}$ mice (Fig. 5D). In line with this observation, CXCL5 levels were similar in total protein DLVP extracts of $Pten/Vdr^{(i)pe-/-}$ and of $Pten^{(i)pe-/-}$ mice 9 months AGI (Fig. EV4C). Importantly, we unraveled a positive association between PECs dissemination into the liver with circulating, but not resident, neutrophil levels (Fig. 5E,F). To gain insight into the role of neutrophils, we targeted the CXCL5/CXCR2 axis involved in neutrophil chemotaxis (Lu et al, 2017) and selectively detected in the neutrophil cluster in our scRNA-seq data (Fig. 5G). To do so, $Pten/Vdr^{(i)pe-/-}$ mice were treated at 9 months AGI for 1 week with the CXCR1/CXCR2 inhibitor, SX-682 (Fig. 5H). Importantly, the

incidence of mouse with infiltrations in liver was reduced in the SX-682-treated group (Fig. 5I). Even though the number of neutrophils in blood analyzed 24 h after the last SX-682 administration was similar between the different treatment groups (Fig. 5J), the proportion of neutrophils in DLVPs was reduced by 2-fold in SX-682-treated animals (Fig. 5K). Note that the frequency of CD8 + / PD1 + T cells in prostates was similar to those of vehicle-treated mice (Fig. 5L). These findings align with previously published data (Lu et al, 2017), showing that PD-1 inhibitors are required to reactivate CD8 + T cell function. Further analysis of the remaining dissemination in the liver from $Pten/Vdr^{(i)pe-/-}$ mice showed that the percentage of F4/80+ or CD3+ cells are not impacted by SX-682 treatment, while that of Ly-6G+ cells is 1.8-fold lower (Fig. 5M,N). In line with these results, immunostaining revealed that the number of cells positive for myeloperoxidase (MPO), a marker of neutrophil activation, was lower in livers of SX-682-treated mice compared to vehicle-treated ones (Fig. 5O). Furthermore, the percentage of PanCK+ cells in the remaining liver infiltrates was 3-fold reduced (Fig. 5P,Q). Thus, neutrophils play a key role to sustain liver infiltrates in $Pten/Vdr^{(i)pe-/-}$ mice and represent a promising therapeutic target to eliminate liver dissemination.

## Discussion and perspectives

In this study, we demonstrate that VDR signaling in PECs prevents PCa progression, by restraining OS and neutrophil recruitment, which we identify as potential biomarkers and/or therapeutic targets for PCa metastasis.

The extensive characterization of $Pten/Vdr^{(i)pe-/-}$ mice unraveled that VDR deficiency does not impact PIN formation, but induces mitochondrial activity and promotes OS that in turn induces 8-OHdG in PECs, a marker of OS-induced DNA damage that correlates with PCa progression (Ohtake et al, 2018; Oh et al, 2016). VDR-loss-increased OS was shown to induce apoptosis in immortalized cancer cell lines (e.g., HaCaT and MCF-7) (Consiglio et al, 2014; Ricca et al, 2018). However, our results show that it does not induce transversion, nor apoptosis in PEC of $Pten/Vdr^{(i)pe-/-}$

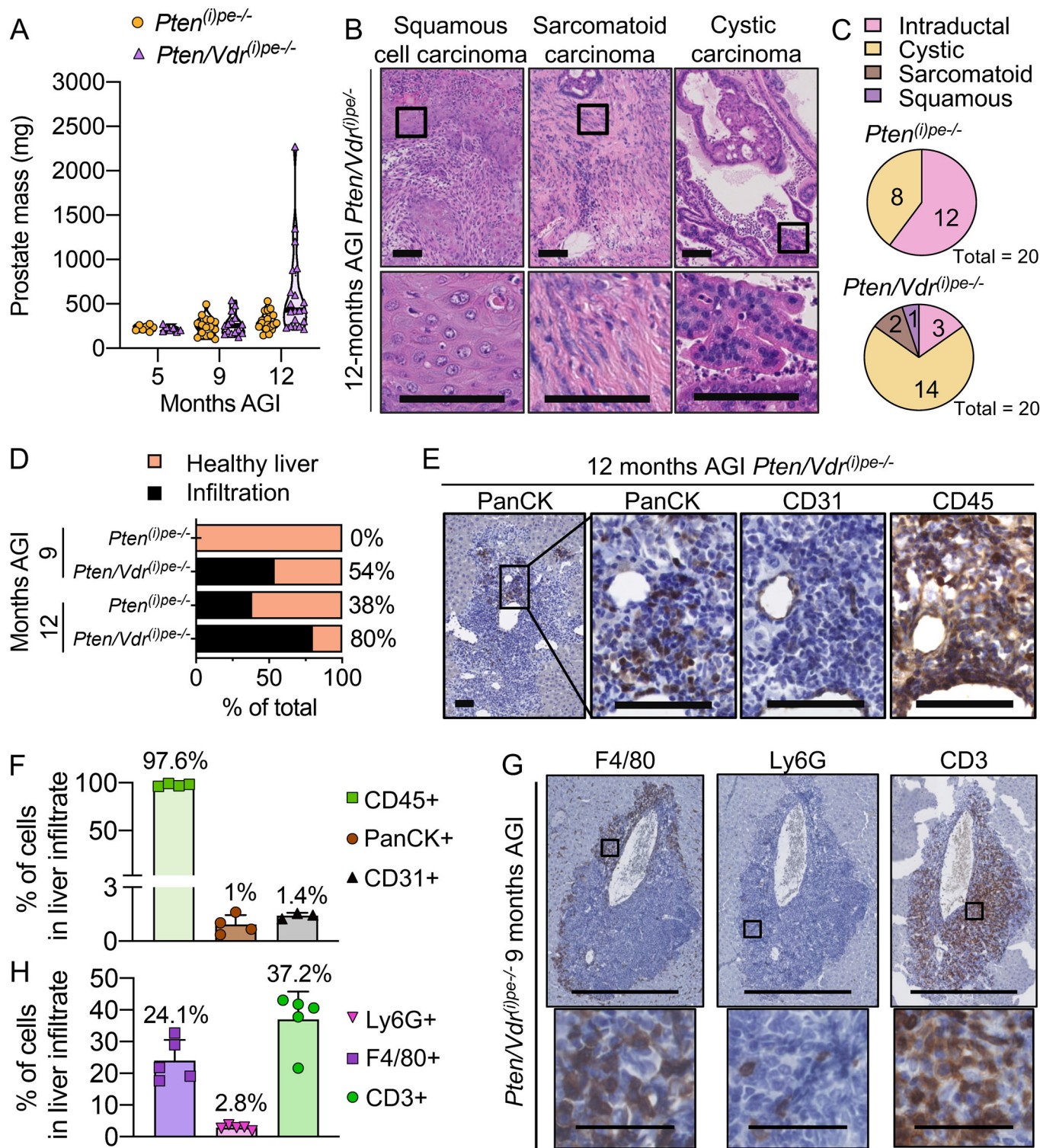

mice. Our data demonstrate that OS in PTEN-deficient PECs promotes cell growth at early PCa stage, as ROS scavenging by NAC reduces PECs proliferation in *Pten/Vdr*[(i)pe-/-] mice. Taken together, these results reveal that VDR plays an instrumental role in PCa progression by reducing PTEN-loss induced OS.

The clinical relevance of vitamin D in PCa was assessed by mining data from a cohort of newly diagnosed French PCa patients with no prior history of vitamin D-calcium supplementation. The analysis revealed a statistically significant correlation between circulating 25(OH) $D_3$ levels and PSA, a predictive marker of PCa progression. Importantly,

**Figure 4. PCa progression and dissemination in *Pten/Vdr*(i)pe-/- mice.**

(A) Prostate mass of *Pten*(i)pe-/- and *Pten/Vdr*(i)pe-/- mice, 5 (*n* = 7), 9 (*n* = 16) and 12 (*n* = 20) months AGI. (B) Representative HE staining of AP sections from *Pten/Vdr*(i)pe-/- mice 12 months AGI. Scale bar = 100 µm (large view) and 50 µm (inset). (C) Proportion of histological subtypes identified by HE staining in AP sections from *Pten*(i)pe-/- and *Pten/Vdr*(i)pe-/- mice 11–12 months AGI. *n* = 20 mice/group. (D) Percentage of *Pten*(i)pe-/- and *Pten/Vdr*(i)pe-/- mice 9 and 12 months AGI with cell dissemination in liver. *n* = 6 mice/group (9 months AGI), 20 mice/group (12 months AGI). Representative immunostaining (E) and quantification (F) of PanCK, CD31 and CD45 positive cells on liver sections from *Pten/Vdr*(i)pe-/- mice 12 months AGI. Data are represented as mean + standard deviation. Scale bar = 500 µm (large view) and 50 µm (zoom). *n* = 4 mice. Representative immunostaining (G) and quantification (H) of F4/80, Ly-6G, and CD3 positive cells on liver sections from *Pten/Vdr*(i)pe-/- mice 9 months AGI. Data are represented as mean + standard deviation. *n* = 5 mice/group. Scale bar = 500 µm (large view) and 50 µm (zoom). Source data are available online for this figure.

the modest coefficient of determination in our study likely reflects the variability in PSA levels among patients, influenced by genetic, metabolic, and environmental factors. In addition, 25(OH)D$_3$ levels in peripheral blood fluctuate in response to seasonal sunlight exposure and dietary intake, whereas PCa develops and progresses over decades. For the present study, this temporal mismatch might limit the explanatory power of a single time-point measurement of vitamin D and PSA levels. Recent studies proposed that the effects of 1,25D$_3$ on prostate cancer are mediated by both its nuclear (VDR) and membrane-associated (PDIA3) receptors (Kermpatsou et al, 2024). In our present study, as VDR levels are increased following PTEN-loss in PEC from both mouse and human, we aimed to determine the role of VDR in PECs on PCa severity. Thus, we undertook further investigation through longitudinal studies, using the *Pten/Vdr*(i)pe-/- mouse model.

VDR was previously shown to reduce cellular senescence in osteoporosis (Xu et al, 2022; Yang et al, 2020). Additionally, our previous work shows that enhanced proliferation induces replicative stress and senescence of PTEN-deficient PECs (Parisotto et al, 2018). Interestingly, VDR-loss enhanced PECs proliferation in *Pten/Vdr*(i)pe-/- mice at 1 month AGI does not accelerate their entry into senescence. However, it impacts the senescence-associated secretory phenotype, as Luminal C cell expression of CXCL5, the main neutrophil chemoattractant, is enhanced in *Pten/Vdr*(i)pe-/- mice. In agreement with this observation, the recruitment of neutrophils is increased in the prostate of these mice at 3 months AGI. Importantly, plasma vitamin D levels in PCa patients negatively correlate with circulating neutrophils. Interestingly, both VDR-loss and OS have been shown to enhance cytokine release (Khandrika et al, 2009; Kuo et al, 2022; Feldman et al, 2014). However, the mechanism underlying enhanced CXCL5 expression in a PTEN and VDR-deficient context remains to be identified.

Progression to mCRPC in patients initially diagnosed with non-metastatic or mHNPC urgently requires biomarkers to improve current treatment strategies. Lymph nodes and bones metastatic sites are considered as classical and indolent, in contrast to visceral metastasis (liver metastasis) and pulmonary metastasis. There is an increased prevalence of liver metastasis in newly diagnosed PCa patients, either synchronous or metachronous, and this localization is predictive of a worse prognosis requiring urgent use of chemotherapy in addition to ADT and ARPIs (Ni et al, 2024; Yeked̈uz et al, 2023). However, the lack of representative preclinical models remains a significant challenge for studying this process. Notably, we demonstrate that PECs from *Pten/Vdr*(i)pe-/- mice disseminate to the liver earlier and with greater frequency than those from *Pten*(i)pe-/- mice, offering a valuable model to investigate the mechanisms driving dissemination. Importantly, liver infiltrations in *Pten/Vdr*(i)pe-/- mice are similar to those recently identified in *Pten/Trp53*(i)pe-/- mice, a known aggressive tumor subtype induced by *Pten* and *Trp53* inactivation (Yanushko et al, 2025). Note that

rare histological subtypes (e.g., sarcomatoid or cystic AdC) observed in AP of *Pten/Vdr*(i)pe-/- mice were previously reported in *Pten/Trp53*(i)pe-/- mice (Yanushko et al, 2025), but their presence does not correlate with cell dissemination. Thus, VDR-loss represents an important event during PCa progression leading to a poor prognosis for liver dissemination.

In previous reports, circulating neutrophils were associated with cell dissemination in mCRPC patients (Autio et al, 2015; Mehra et al, 2016). Here we show that they correlate with liver infiltration incidence in *Pten/Vdr*(i)pe-/- mice and represent a potential diagnosis tool for liver metastasis in PCa. It is noteworthy that liver metastasis usually occurs at the end-stage of mCRPC, when PSA levels are sometimes low, reflecting plasticity from adenocarcinoma to neuro-endocrine or mesenchymal/stem PCa. At the time of recruitment, none of the cohort patients presented with liver metastasis. Their follow-up, notably for those with a high PSA and low 25(OH)D$_3$ levels, will be informative to gain insight into the protective role of vitamin D levels on tumor spread.

Our findings on liver infiltrations in *Pten/Vdr*(i)pe-/- mice align with a previously published study (Hawley et al, 2023), reporting that liver metastases in mHNPC patients are characterized by high immune cell infiltration associated with treatment response. Note that in this study neutrophil infiltration was also reported in liver metastases of mHNPC patients, however, their role in tumor cell dissemination and treatment response remains unexplored. Here, we demonstrate that targeting neutrophil chemotaxis by a one-week treatment with SX-682 impacts liver metastasis in mHNPC. Given the short treatment time, the drug is likely targeting the existing niches rather than de novo ones. Indeed, we observed the reduction of liver-infiltrating neutrophils and PanCK+ cells in the remaining niches. Despite decreased levels of prostate-infiltrating neutrophils by SX-682 treatment, cytotoxic T cells remain exhausted. This finding indicates that the reduced liver metastasis is not associated with T cell activity in situ, but with the number of circulating neutrophils.

Taken together, this work demonstrates how VDR signaling via ROS scavenging in PECs limits PIN progression. In addition, we provide evidence that reducing neutrophil chemotaxis represents a promising therapeutic target to prevent and eliminate the metastatic spread in PCa.

## Methods

### Patient data

Informed consent was obtained from all human subjects and the experiments conformed to the principles set out in the WMA Declaration of Helsinki and the Department of Health and Human

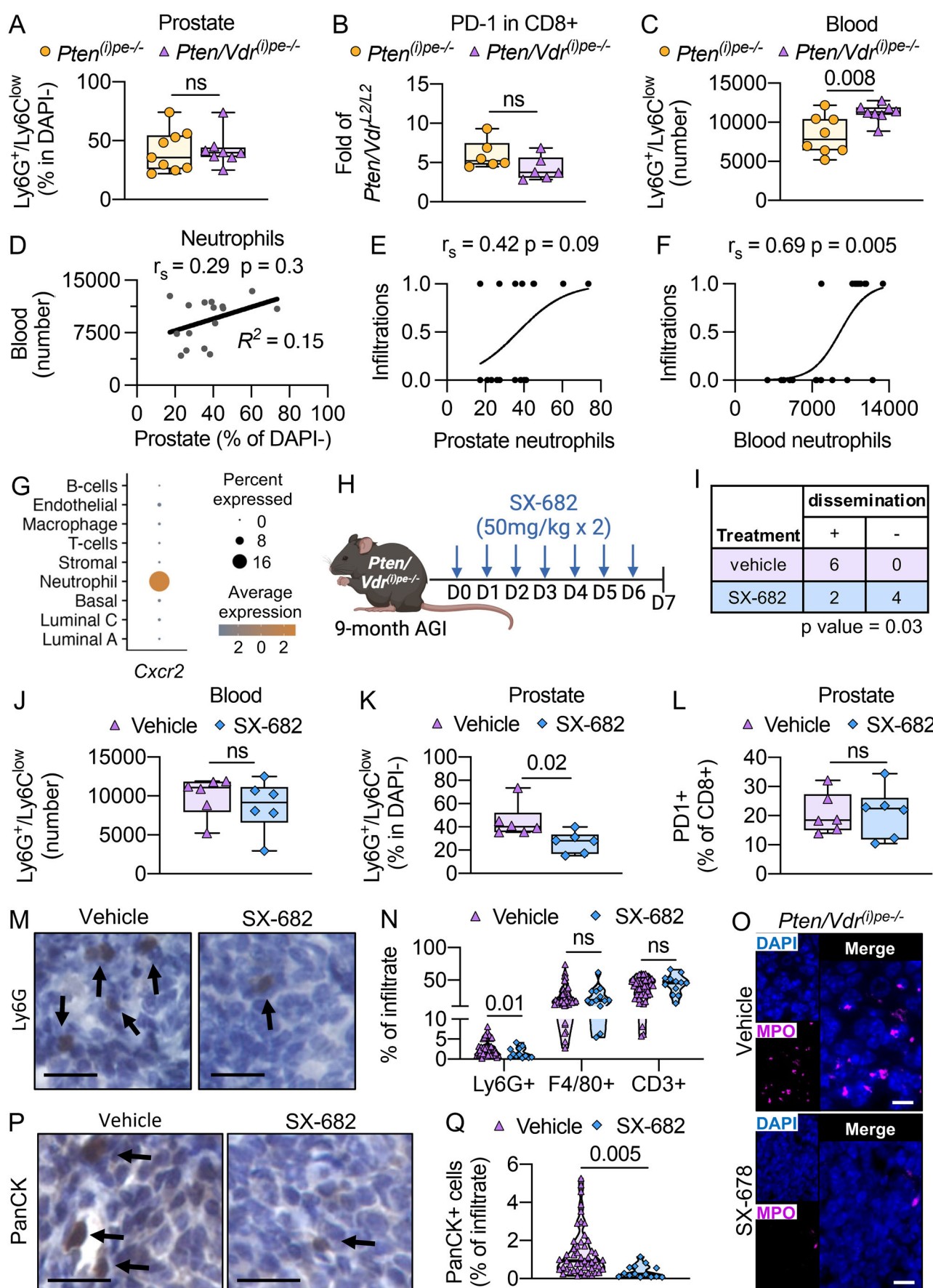

◄ **Figure 5. Neutrophils involvement in PECs metastasis to the liver.**

(A) Flow cytometry analysis of alive (DAPI-) Ly-6G$^{high}$/Ly-6C$^{low}$ neutrophils quantified in DLVPs of $Pten^{(i)pe-/-}$ and $Pten/Vdr^{(i)pe-/-}$ mice 9–11 months AGI. The boxes extend from the 25th to 75th percentiles, the lines represent the median, and the whiskers go down to the smallest up to the largest value. $n = 8$–9 mice/group. Student's $t$-test. ns, $p \geq 0.05$. (B) Flow cytometry analysis of alive (DAPI-) CD8 + /PD-1 + T cells quantified in DLVP of $Pten^{(i)pe-/-}$ and $Pten/Vdr^{(i)pe-/-}$ mice 9 months AGI, represented as a fold of age-matched $Pten/Vdr^{L2/L2}$ mice. The boxes extend from the 25th to 75th percentiles, the lines represent the median, and the whiskers go down to the smallest up to the largest value. $n = 6$ mice/group. The $p$ value determined by Student's $t$-test is indicated. ns, $p \geq 0.05$. (C) Number of neutrophils in blood from $Pten^{(i)pe-/-}$ and $Pten/Vdr^{(i)pe-/-}$ mice 9–11 months AGI. The boxes extend from the 25th to 75th percentiles, the lines represent the median, and the whiskers go down to the smallest up to the largest value. $n = 8$ mice/group. The $p$ value determined by Student's $t$-test is indicated. (D) Scatter plot with estimated linear regression line (black) for circulating neutrophils and the prostate-infiltrated neutrophils in $Pten/Vdr^{(i)pe-/-}$ mice 9 months AGI. $n = 16$ mice. Spearman correlation coefficient ($r_s$), $p$ value and coefficient of determination ($R^2$) are indicated. Scatter plot with estimated simple logistic regression S-curve (black) showing the relationship between liver infiltration incidence and prostate-infiltrated neutrophils (E) or circulating neutrophils (F) in $Pten/Vdr^{(i)pe-/-}$ mice 9 months AGI. $n = 16$ mice. Spearman correlation coefficient ($r_s$) and $p$ value are indicated. (G) Dot plot representing the expression of $Cxcr2$ in the various cellular clusters identified by scRNA-seq. (H) Schematic representation of SX-682 administration to $Pten/Vdr^{(i)pe-/-}$ mice. (I) Number of $Pten/Vdr^{(i)pe-/-}$ mice 9 months AGI with liver metastasis after a 1-week SX-682 or vehicle treatment. $n = 6$ mice/group. The $p$ value determined by one-sided Fisher's exact test is indicated. (J) Number of neutrophils in blood from $Pten/Vdr^{(i)pe-/-}$ mice 9 months AGI treated with vehicle or SX-682 for 1 week. The boxes extend from the 25th to 75th percentiles, the lines represent the median, and the whiskers go down to the smallest up to the largest value. $n = 6$ mice/group. The $p$ value determined by Student's $t$-test is indicated. ns, $p \geq 0.05$. (K) Flow cytometry analysis of alive (DAPI-) Ly-6G$^{high}$/Ly-6C$^{low}$ neutrophils in DLVPs of $Pten/Vdr^{(i)pe-/-}$ mice 9 months AGI treated with vehicle or SX-682 for 1 week. The boxes extend from the 25th to 75th percentiles, the lines represent the median, and the whiskers go down to the smallest up to the largest value. $n = 6$ mice/group. The $p$ value determined by Student's $t$-test is indicated. (L) Flow cytometry analysis of CD8 + /PD-1 + T cells quantified in DLVP of $Pten/Vdr^{(i)pe-/-}$ mice 9 months AGI treated with vehicle or SX-682 for 1 week. The boxes extend from the 25th to 75th percentiles, the lines represent the median, and the whiskers go down to the smallest up to the largest value. $n = 6$ mice/group. The $p$ value determined by Student's $t$-test is indicated. ns, $p \geq 0.05$. (M) Representative immunostaining of Ly-6G on liver sections from $Pten/Vdr^{(i)pe-/-}$ mice 9 months AGI. $n = 5$ mice/group. Scale bar = 50 μm. Arrows point to positive cells. (N) Quantification of Ly-6G +, F4/80 +, CD3+ cells per infiltration performed on immunostained liver sections from vehicle- and SX-682-treated $Pten/Vdr^{(i)pe-/-}$ mice 9 months AGI. $n = 4$ (vehicle-treated) and 2 (SX-682-treated) mice. (O) Representative immunostaining of MPO on liver sections from vehicle- and SX-682-treated $Pten/Vdr^{(i)pe-/-}$ mice 9 months AGI. Scale bar = 10 μm. $n = 4$ (vehicle-treated) and 2 (SX-682-treated) mice. MPO (magenta) and DAPI (blue). Representative immunostaining (P) and quantification (Q) of PanCK+ cells per infiltration performed on liver sections from vehicle- and SX-682-treated $Pten/Vdr^{(i)pe-/-}$ mice, 9 months AGI. $n = 4$ (vehicle-treated) 2 (SX-682-treated) mice. Scale bar = 50 μm. Arrows point to positive cells. Source data are available online for this figure.

Services Belmont Report. Data from 198 patients from the HormOS cohort were collected (patients from two academic onco-urological centers from Paris University Hospitals initiating or resuming ADT +/− ARPIs +/− radiation therapy and referred for a biological and radiographic rheumatological assessment to evaluate bone mineral density). Among them, we analyzed only patients with a newly diagnosed PCa either localized or metastatic, excluding patients with recurrence, who had already received hormone therapy. Note that the patients were treated for less than 19 days in median before 25(OH)D$_3$ blood analysis. A complete peripheral blood sample was performed with specific analysis on vitamin D levels and numeration. The lymphocyte content was estimated by the number of total leukocytes minus the polynuclear neutrophils. Biological, histological, and clinical data of the patients were prospectively collected on a Redcap sheet, after local Comité de Protection des Personnes (CPP) agreement (CLEP Decision N°: AAA-2022-08028). The ethnic origin was not available due to ethical and reglementary restrictions.

### Mice

$Pten^{(i)pe-/-}$ mice on a pure C57Bl/6 background have been generated as described (Ratnacaram et al, 2008). $Pten/Vdr^{(i)pe-/-}$ mice on a pure C57Bl/6 background were generated by intercrossing mice expressing the tamoxifen-dependent CreER$^{T2}$ recombinase under the control of the human PSA promoter in luminal cells of the prostatic epithelium and $Pten$ exons flanked by LoxP sites (PSA-CreER$^{T2(tg/0)}$/Pten$^{L2/L2}$) with mice in which VDR alleles are floxed ($Vdr^{L2/L2}$) (Yamamoto et al, 2013). Several intercrosses generated PSA-CreER$^{T2(tg/0)}$/Pten$^{L2/L2}$/Vdr$^{L2/L2}$ and PSA-CreER$^{T2(0/0)}$/Pten$^{L2/L2}$/Vdr$^{L2/L2}$ littermates. Gene inactivation was induced in 8/10-week-old mice by intraperitoneal tamoxifen injections (1 mg per mouse) for five consecutive days. Mouse breeding and maintenance were carried out in the accredited animal facility of the Institut de génétique et de biologie moléculaire et cellulaire (IGBMC), in compliance with French and European Union regulations on the use of laboratory animals for research. Only male mice were analyzed, as prostate cancer modeling is relevant only in animals of this sex. Investigators were not blinded to animals' genotypes. At the designated timepoint post-tamoxifen administration, mice were randomly assigned to treatment groups. NAC (Sigma-Aldrich, A0737) (2%) was dissolved in drinking water. NAC treatment started 2 days post tamoxifen administration for 4 weeks. SX-682 (MedChemExpress; HY-119339) was diluted in corn oil and orally administered twice a day at 50 mg/kg for 1 week. All animal experimental protocols followed ARRIVE guidelines and were approved by the Ethical Committee Com'Eth (Comité d'Ethique and the French Ministry of Higher Education and Research (#APAFIS #37935-2022070811077390 v4).

### Genotyping

Genomic DNA was isolated from mouse biopsies with the DirectPCR extraction kit (102-T; Viagen), and the $Pten$ alleles were amplified by polymerase chain reaction (PCR) as described (Ratnacaram et al, 2008). For $Vdr$ alleles detection, V1 sense (5'-GTCTTCTGACTCCCACAAGTGTACC-3') and V2 antisense primer (5'-CTGTTGATGGACAGGAACACACAGC-3') were used to amplify the 280-bp $Vdr$ WT or 350-bp $Vdr$ L2 DNA segments, and sense V1 and antisense primer V3 (5'-CTTTGTACTACCAGGCT-GAGCTTCG-3') were used to amplify the 320-bp $Vdr$ L- DNA segment.

### Cell line

The human non-malignant prostatic epithelial cell line RWPE-1 (CVCL_3791, ATCC, Cat# CRL-1160), tested for mycoplasma contamination, was cultured in Keratinocyte Serum-Free Medium (KSFM; Life Technologies, Cat# 17005-042) supplemented with 5 ng/mL human recombinant epidermal growth factor (Bioscience, Cat# CC-4107), 0.05 mg/mL bovine pituitary extract (Gibco, Cat#

13028014), and 40 µg/mL gentamicin (Gibco, Cat# 15750060). Cells were maintained at 37 °C in a humidified incubator with 5% $CO_2$. For experimental treatments, cells were seeded at approximately 70% confluence and incubated with 1 mM NAC, dissolved in sterile water, for 48 h.

### Western blotting

Protein isolation was performed on tissues and cells lysed in radioimmunoprecipitation assay buffer (RIPA) [50 mM Tris pH 7.5, 1% Nonident P40, 0.5% sodium deoxycholate, 0.1% SDS, 150 mM NaCl, 5 mM EDTA, 1 mM PMSF, and phosphatase and protease inhibitor cocktails (PhosphoStop and Complete-Mini EDTA free; Roche)]. Tissue lysis was performed on ice, using a glass Dounce Homogeniser Pestle B. Lysates were cleared by centrifugation at $10,000 \times g$ for 10 min (min) at 4 °C. Supernatant protein concentrations were determined by colorimetric assay (Abcam, ab119216) according to the manufacturer's instructions. Equal protein amounts were separated by electrophoresis under denaturing conditions in 10% sodium dodecyl sulfate (SDS)–polyacrylamide gel and transferred to nitrocellulose membranes (Trans-blot turbo transfer system, Bio-Rad), following the manufacturer's protocol. Membranes were blocked by 5% nonfat dry milk in 10 mM Tris pH 7.4, 0.05% Tween-20 (TBST) for 1 h (h) at room temperature (RT). Membranes were cut considering the target molecular weight using the PageRuler™ Plus Prestained Protein Ladder, 10 to 250 kDa (11832124, ThermoScientific) as landmark, and incubated overnight with one of the primary antibodies [rabbit anti-VDR (Cell Signaling Technology Cat# 12550, RRID:AB_2637002; 1:500 dilution), mouse anti-β-Actin (Sigma-Aldrich Cat# A1978, RRID:AB_476692; 1:1000 dilution), rabbit anti-PTEN (Cell Signaling Technology Cat# 9559S, RRID:AB_390810; 1:500 dilution), rabbit anti-GAPDH (Cell Signaling Technology Cat# 2118S, RRID:AB_561053; dilution 1:10,000)] diluted in 5% bovine serum albumin (BSA) in TBST at 4 °C. After washes in TBST, anti-mouse immunoglobulin G (IgG) (Cell Signaling Technology Cat# 7076S) or anti-rabbit IgG (Cell Signaling Technology Cat# 7074S) horseradish peroxidase (HRP)–linked antibodies diluted at 1:10,000 in 5% nonfat dry milk in TBST was added for 1 h at RT, and membranes were incubated with an enhanced chemiluminescence detection system (ECLplus, GE Healthcare) and imaged [ImageQuant LAS 4000 biomolecular imager (GE Healthcare)]. Immunodetected proteins were quantified with Fiji/ImageJ software (Schindelin et al, 2012).

### Histological examination

Tissues were fixed in 10% formalin overnight. Samples were embedded in paraffin, and 5 µm serial sections were deparaffinized and stained with hematoxylin and eosin (HE), as per standard protocols. Histological subtypes were identified by Véronique Lindner (Pathology department, CHU of Strasbourg - Hautepierre Hospital). Histological scoring was performed on HE-stained and α-SMA-immunostained prostatic sections. Gland area was quantified on HE-stained prostatic sections using QuPath software (Bankhead et al, 2017).

### Immunohistochemistry

Five µm sections of paraffin embedded tissues were deparaffinized and antigen retrieval was performed using SignalStain® Citrate Unmasking Solution (1X) (Cell Signaling Technology Cat# 14746) in a pressure cooker for 20 min. Sections were treated with 3% $H_2O_2$. Non-specific antigens were blocked by 5% nonfat dry milk in TBST for 1 h at RT. Immunostaining was performed by an overnight incubation at 4 °C with primary antibodies [rabbit anti-p-Akt Ser473 (Cell Signaling Technology Cat# 4060, RRID:AB_2315049; 1:200 dilution), rabbit anti-α-SMA (Cell Signaling Technology Cat# 19245, RRID:AB_2734735; 1:200 dilution), mouse anti-8-OHdG (Abcam Cat# ab48508, RRID:AB_867461; 1:100 dilution), rabbit anti-Ki-67 (Thermo Fisher Scientific Cat# MA5-14520, RRID:AB_10979488; 1:200 dilution), rabbit anti-PanCK (Proteintech Cat# 26411-1-AP, RRID:AB_2880505; 1:500 dilution), rabbit anti-CD45 (Abcam Cat# ab10558, RRID:AB_442810; 1:600 dilution), goat anti-CD31 (R and D Systems Cat# AF3628, RRID:AB_2161028; 1:200 dilution), rabbit anti-F4/80 (Cell Signaling Technology Cat# 70076, RRID:AB_2799771; 1:200 dilution), rabbit anti-CD3 (Abcam Cat# ab16669, RRID:AB_443425; 1:200 dilution), rabbit anti-CK19 (Abcam Cat# ab133496, RRID:AB_11155282; 1:200 dilution)]. Ready to use anti-rabbit (Cell Signaling Technology Cat# 8114) or anti-mouse (Cell Signaling Technology Cat# 8125) Boost IHC Detection HRP-conjugated secondary antibody was added for 30 min at RT. Antigen detection was performed with 3,3′-diaminobenzidine (DAB) substrate kit (Cell Signaling Technology Cat# 8059). Slides were stained by hematoxylin, as per standard protocols. Sections processed in the absence of primary antibodies were used as negative control.

### Immunofluorescence

Five µm sections of paraffin embedded tissues were deparaffinized according to standard protocols and antigen retrieval was performed using SignalStain® Citrate Unmasking Solution (1X) (Cell Signaling Technology Cat# 14746) in a pressure cooker for 20 min. For in vitro immunofluorescence assays, RWPE-1 cells were cultured on Lab-Tek II chamber slides (Thermo Fisher Scientific, Cat# 154534) and fixed with 4% paraformaldehyde (PFA) for 15 min at room temperature. Cells were then permeabilized with 0.2% Triton X-100 (Sigma-Aldrich, Cat# 108603) in PBS for 5 min and blocked with 5% bovine serum albumin (BSA) in PBS for 1 h. Immunostaining was performed by overnight incubation at 4 °C with rat anti-Ly-6G antibody (BioLegend Cat# 127602, RRID: AB_1089180; 1/200), rabbit anti-MPO (Cell Signaling Technology Cat# 14569, RRID:AB_2798516; 1:200 dilution), rabbit anti-Ki-67 (Thermo Fisher Scientific Cat# MA5-14520, RRID:AB_10979488). After washes, goat anti-rat IgG (H + L) Highly Cross-Adsorbed Alexa Fluor Plus 555-coupled secondary antibody (Thermo Fisher Scientific Cat# A48263, RRID:AB_2896332) or goat anti-rabbit IgG (H + L) Highly Cross-Adsorbed Alexa Fluor Plus 555-coupled secondary antibody (Thermo Fisher Scientific Cat# A-21428, RRID:AB_2535849) was added at a dilution of 1:400 in PBS for 1 h at RT. Sections were mounted in Fluoromount-G Mounting Medium with 4′,6-diamidino-2-phenylindole (DAPI, Thermo Fisher Scientific Cat# 00-4959-52) before microscopic acquisition.

### Gene silencing

RWPE-1 cells were transfected with 25 nM siRNA targeting human PTEN (Dharmacon, Cat# L-003023-00-0005) and/or 50 nM siRNA targeting human VDR (Dharmacon, Cat# L-003448-00-0005), or

75 nM non-targeting control siRNA (Dharmacon, Cat# D-001810-10-05) using Lipofectamine RNAiMAX reagent (Thermo Fisher Scientific, Cat# 13778150) according to the manufacturer's instructions. Briefly, cells were seeded in serum-free medium and grown to approximately 80% confluence prior to transfection. Following 48 h of transfection, cells were subjected to the indicated treatments and analyses.

### Mitochondrial activity

Mitochondrial function was assessed in RWPE-1 cells using a Seahorse XF96 Extracellular Flux Analyzer (Agilent Technologies, UK) according to the manufacturer's protocol. Briefly, RWPE-1 cells were seeded in XFe96 cell culture microplates (Agilent Technologies, Cat# 103793-100) and transfected with siRNA targeting PTEN and/or VDR, or with a non-targeting control siRNA. Prior to the assay, cells were incubated for 1 h at 37 °C in a $CO_2$-free incubator with Seahorse XF assay medium (Agilent Technologies Cat# 103680-100) supplemented with 25 mM glucose (Gibco Cat# A2494001), 1 mM sodium pyruvate (Gibco Cat# 11360070), and 2 mM glutamine (Gibco Cat# A2916801). Oxygen consumption rate (OCR; pmol/min) was measured under basal conditions and after sequential injections of oligomycin (ATP synthase inhibitor; 1 μM, Sigma-Aldrich Cat# 75351), carbonyl cyanide m-chlorophenylhydrazone (FCCP; mitochondrial uncoupler; 2 μM, Sigma-Aldrich Cat# 215911), and a combination of rotenone (complex I inhibitor; 0.5 μM, Sigma-Aldrich Cat# R8875) and antimycin A (complex III inhibitor; 0.5 μM, Sigma-Aldrich Cat# A8674). OCR values were normalized to total protein content in each well, quantified using the Bradford Protein Assay (Abcam Cat# ab119216). Basal and maximal respiration rates were determined using the Seahorse Wave software according to the manufacturer's instructions.

### TUNEL assay

Five μm sections of paraffin-embedded tissues were deparaffinized according to standard protocols and analyzed with the In Situ Cell Death Detection Kit, Fluorescein (Roche Cat# 11684795910) following the manufacturer's instructions. Positive control corresponds to the tissue treated with DNAse I (Roche Cat# 471672800) for 10 min at RT. Fluoromount-G Mounting Medium with DAPI (Thermo Fisher Scientific Cat# 00-4959-52) was applied before microscopic acquisition.

### SA-β-gal staining

Frozen sections (10 μm) prepared from optimal cutting temperature (OCT) compound-embedded prostates were analyzed with the Senescence β-Galactosidase Staining Kit (Cell Signaling Technology Cat# 9860) following the manufacturer's instructions. Sections were counterstained with hematoxylin and mounted before image acquisition.

### Image acquisition

Brightfield images were scanned using 20× magnification with a NanoZoomer S210 Digital slide scanner (Hamamatsu). The NDP.view2 software (Hamamatsu) was used for image editing. Ki-67- and 8-OHdG-positive cells were quantified with the QuPath v3 software (Bankhead et al, 2017). Fluorescence was acquired using an upright motorized microscope (Leica DM 4000 B) with 40× N PLAN (NA, 0.65) objective, the CoolSnap CF Color camera (Photometrics) and the Micro-Manager software. The Fiji/ImageJ software was used for image editing (Schindelin et al, 2012).

### CXCL5 levels

CXCL5 levels contained in twenty micrograms of DLVP protein extract were determined using the Mouse LIX ELISA Kit (CXCL5) (Abcam Cat# ab264611), according to manufacturer's protocol.

### Tissue dissociation

Prostate dissociation into single cells was performed as described (Abu El Maaty et al, 2021). Briefly, DLVPs were mechanically minced, and then enzymatically dissociated in DMEM [glucose (4.5 g/liter), GlutaMAX, 10% FCS, 1% P/S] containing collagenase type I (1 mg/ml) (Thermo Fisher Scientific Cat# 17018029), with mild agitation for 40 min. Subsequently, trypsin/0.05% EDTA (Thermo Fisher Scientific Cat# 25300) was added to the dissociated tissues for 5 min at 37 °C. Cells were then resuspended in DMEM containing 500 U of RNase-free DNase I (Roche Cat# 04716728001). Cellular clumps were dissociated by passing the suspension at least 5 times through a 20-gauge needle and once through a 40-μm cell strainer (Corning Cat# 352340). The cell suspension was washed in FACS buffer (BSA 5%, EDTA 2 mM in PBS).

### DLVP immunophenotyping

Single cells isolated from a prostate were incubated with an anti-CD16/32 antibody (BD Pharmingen Cat# 553141 RRID: AB_394656; 1:50 dilution) for 15 min on ice, then washed in FACS buffer and stained with an antibody cocktail for 15 min on ice. The antibody cocktail for neutrophil detection included FITC-coupled anti-CD45 (BioLegend Cat# 103107, RRID: AB_312972), PerCP–Cy5.5-coupled anti-CD11b (Thermo Fisher Scientific Cat# 45-0112-82, RRID: AB_953558), BV785-coupled anti-Ly-6G (BioLegend Cat# 127645, RRID: AB_2566317), PE-CF594-coupled anti-Ly-6C (BD Horizon Cat# 562728, RRID: AB_2737749). All antibodies were diluted 1/100. The antibody cocktail for T cells detection included FITC-coupled anti-CD45 (BioLegend Cat# 103107, RRID: AB_312972; 1:100 dilution), PerCP-Cy5.5-coupled anti-CD3 (BioLegend Cat# 100218, RRID: AB_1595492; 1:25 dilution), Brilliant Violet 785-coupled anti-CD8 (BioLegend Cat# 344739, RRID: AB_2566202; 1:50 dilution), APC-H7 anti-CD4 (BD Pharmingen Cat# 560158, RRID: AB_1645478; 1:100 dilution), APC-coupled anti-PD-1 (BioLegend Cat# 379207, RRID: AB_2922606; 1 μL/sample). Dead cells were stained with DAPI. Samples were analyzed by a BD Fortesa flow cytometer and the FlowJo v10 Software (RRID:SCR_008520).

### Blood immunophenotyping

Blood mouse was collected in 1.3 ml Lithium Heparin Tubes (ThermoFischer) and incubated with an equal volume of antibody cocktail for neutrophil detection for 30 min at RT in the dark. FACS Lysing solution (BD FACS Cat# 349202) was then added to the sample solution. After 5 min, the sample was directly analyzed on a BD Fortesa flow cytometer and using FlowJo v10 Software. The frequency of circulating neutrophils was determined as Ly-6G$^+$/Ly-6C$^{low}$ cells count per 120 s of flow record.

### Luminal cell sorting

Single cell suspension from DLVP from 3 mice per condition was stained with an antibody cocktail for luminal cell detection

(Sackmann Sala et al, 2017), including FITC-coupled lineage 'Lin' antibodies [anti-CD31 (BioLegend Cat# 103101, RRID: AB_312966), anti-CD45 (BioLegend Cat# 103107, RRID: AB_312972), and anti-Ter119 (BioLegend Cat# 116206, RRID: AB_313706); all at 1:250 dilution], PE-coupled anti-CD49f (BioLegend Cat# 313611, RRID: AB_893374; 1:25 dilution), and APC-coupled anti-Sca-1 (BioLegend Cat# 160904, RRID: AB_2910333; 1:75 dilution). Dead cells were stained with DAPI. DAPI- luminal cells (Fig. EV2D,E) were isolated with a BD FACS Aria Fusion flow cytometer.

### Transcriptomic analysis

Total RNA was isolated from FACS-isolated luminal cells using TRI Reagent (Molecular Research Center, Inc.) and purified using RNeasy Kits (Qiagen). cDNA libraries were generated from RNA with an integrity number >8 using the SMART-Seq v4 Ultra Low input RNA kit (Takara Bio Cat# 634890), according to the manufacturer's instructions, quantified and checked for quality using capillary electrophoresis. Fifty base pair single-read sequencing was performed on a NextSeq 2000 (Illumina) following the manufacturer's instructions. Reads preprocessing steps were performed using cutadapt 1.10, reads were mapped onto the mm10 assembly of Mus musculus genome using STAR version 2.5.3a. Gene expression quantification was performed from uniquely aligned reads using htseq-count version 0.6.1p1, with annotations from Ensembl version 108 (RRID:SCR_002344) and "union" mode. Only non-ambiguously assigned reads were retained for further analyses. Read counts were normalized across libraries as described (Anders and Huber, 2010). Comparisons of the transcripts were performed on R v4.2.2 and the DESeq2 Bioconductor library (DESeq2 v1.38.3, RRID:SCR_000154) as described (Love et al, 2014). Resulting *p*-values were further adjusted for multiple testing. Genes were considered differentially expressed when the *p*-value was lower than 0.05 and the log2 Fold-change >0.3. Heatmaps were performed with pheatmap v1.0.12 (RRID:SCR_016418). KEGG (RRID:SCR_012773) annotations were performed using ClusterProlifer v4.6.0 and DOSE v3.24.2 as described (Yu et al, 2015). Transversion events were identified from the RNA-seq data following GATK Germline short variant discovery Best Practice Workflow with GATK v4.6.1.0. Importantly, Ensembl's release 104 of Mus musculus genome (GRCm39) and SNP known sites were used as reference, and variants were filtered with the function VariantFiltration applying the filters $QD < 20.0$, $FS > 60.0$, $SOR > 3.0$, $DP < 10$ and $MQ < 50.0$. Plots summarizing the number of transversion in samples were created from vcftools (v0.1.16) --TsTv-summary files.

### scRNA-seq and data analysis

After prostate dissociation, alive (DAPI-) cells were sorted with a BD FACS Aria Fusion flow cytometer. Cell number and viability were determined by a trypan blue exclusion assay on a Neubauer chamber. Samples containing >95% viable cells were processed in parallel on the Chromium Controller from 10X Genomics (Leiden, The Netherlands). Sixteen thousand cells were loaded per well in nanoliter-scale Gel Beads-in-Emulsion (GEMs). Single-cell 3′ mRNA-seq library was generated according to Chromium Single Cell 3′ Reagent Kits User Guide (v2 Chemistry) from 10X Genomics (reference CG00052 Rev E). Briefly, GEMs were generated by combining barcoded gel beads, a reverse transcriptase

### The paper explained

#### Problem
Prostate cancer is life-threatening when it metastasizes, particularly to organs such as the liver. However, diagnosis tools to predict the progression of localized tumor into metastatic disease are in demand. Although low vitamin D levels have been associated to poorer outcomes in several studies, the impact of vitamin D signaling in prostatic epithelial cells on tumor progression remains poorly understood.

#### Results
By analyzing clinical data from a French cohort of newly diagnosed prostate cancer patients, we have shown that those with lower circulating vitamin D levels have higher levels of prostate-specific antigen (PSA), a biomarker associated with tumor burden. Then, we leveraged relevant mouse models of prostate cancer in which the vitamin D receptor (VDR) and/or the tumor suppressor PTEN were inactivated in prostate epithelial cells to investigate the role of vitamin D signaling in PCa. Loss of VDR during the tumor onset increased cellular oxidative stress, which promoted tumor cell proliferation. In addition, tumors lacking vitamin D signaling in epithelial cells had higher neutrophil infiltration and a potent immunosuppressive microenvironment. Over time, these mice developed more aggressive tumors with enhanced cancer cell dissemination to the liver. Importantly, the levels of circulating neutrophils in mice were associated with liver spread, and treatment with a pharmacological agent targeting neutrophil chemotaxis removed metastatic niches.

#### Impact
This study reveals a previously unrecognized role for epithelial vitamin D signaling in restraining prostate cancer progression by limiting oxidative stress and neutrophil-driven metastatic niches. Importantly, it identifies circulating neutrophils as a potential biomarker of metastatic risk and highlights neutrophil recruitment pathways as promising therapeutic targets to prevent or limit cancer cell dissemination. Together, our findings have direct translational implications for patient stratification and therapeutic intervention.

master mix containing cells, and partitioning oil onto Chromium Chip A. Following full-length complementary DNA (cDNA) synthesis and barcoding from polyadenylated mRNA, GEMs were broken and pooled before cDNA amplification by 10 PCR cycles. After enzymatic fragmentation and size selection, sequencing libraries were constructed by adding P5 and P7 primers (Illumina, Paris, France) as well as sample index via end repair, A tailing, adaptor ligation, and PCR amplification with 12 cycles. Library quantification and quality control were performed using Bioanalyzer 2100 (Agilent Technologies, Santa Clara, CA). Libraries were then sequenced on Illumina HiSeq 4000 as 100-base paired-end reads. Image analysis, base calling, and demultiplexing were performed using RTA 2.7.7 and Cell Ranger 3.0.1 mkfastq. Alignment, barcode, and UMI filtering and counting were performed using Cell Ranger v3.0.1 count and mouse reference 3.0.0 (mm10 and Ensembl release 93). The Read10X function of the Seurat v 4 (Stuart et al, 2019) R v 4.0.2 was used to read the output of the Cell Ranger pipeline and obtain a matrix of the number of reads of each gene detected in each cell. Genes expressed in less than 10 cells were excluded. Cells from the different samples with more than 100 and less than 5000 expressed genes and with lower than 20% mitochondrial genes were analyzed, as previously described (Abu El Maaty et al, 2021).

### Data analysis

No statistical method was used to estimate the sample size. No inclusion/exclusion criteria, and no method of randomization were used in this study. No blinding was used for animal studies. Data are represented as mean +/− standard deviation (SD), box, scatter or violin plots. Statistical comparisons were performed using GraphPad Software Prism 10 (RRID:SCR_002798) by Student's t-test, one or two-way ANOVA followed by a post-hoc analysis (Dunnett's or Tukey's post hoc test, respectively), correlation or simple logistic regression analysis, or Fisher's exact test using GraphPad Prism v10. Multivariate linear regression was performed using the *metan* package and the function *corr_coef*. For statistically significant data *p* values were indicated in the figures.

## Data availability

The datasets and computer code produced in this study are available in the following databases: RNAseq data: Gene Expression Omnibus GSE281118. Single cell data: Gene Expression Omnibus GSE281153. Single cell processed data: web application (https://provitd.github.io). Computer scripts: GitHub (https://github.com/provitd/ProCaVit_2026). Imaging datasets: Image Data Resource https://doi.org/10.6019/S-BIAD2776 (https://www.ebi.ac.uk/biostudies/bioimages/studies/S-BIAD2776).

The source data of this paper are collected in the following database record: biostudies:S-SCDT-10_1038-S44321-026-00417-5.

## Peer review information

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

## Acknowledgements

We thank T. Mak and A. Suzuki for floxed *Pten* mice, and S. Kato for floxed *Vdr* mice. We are grateful to Régis Lutzing, Marie Cerciat, Beatriz German-Falcon, Marina Valente-Borroso, Benoit Marteyn, Nicolas Le May, Celine Keime, Christelle Thibault Carpentier, Mohamed Abu el Maaty, the IGBMC animal facility, cell culture and histology service, the flow cytometry facility members Claudine Ebel and Muriel Phillips; GenomEast, a member of the "France Génomique" consortium (ANR-10-INBS-0009); and IGBMC imaging center, a member of the national infrastructure France-BioImaging supported by the French National Research Agency (ANR-10-INBS-04), for excellent assistance. We appreciate the support of Abigail Samuelsen for proofreading and acknowledge the contribution of the patients. This work was funded by Institut National du Cancer (INCa_18498) to GL and SH, by the Association pour la Recherche contre les Tumeurs Prostatiques (ARTP) and La Ligue Contre le Cancer to GL. KL was supported by the Ministry of Higher Education and Research and by the Fondation ARC pour la Recherche sur le Cancer. DY was supported by the Ministry of Higher Education and Research and by the Fondation pour la Recherche Medicale. VF was supported by the INSERM/Pfizer fellowship. The research was also funded by the French National Research Agency (ANR) through the Programme d'Investissement d'Avenir under contract ANR-10-LABX-0030-INRT grant under the frame programme Investissement d'Avenir ANR-10-IDEX-0002-02, as well as by the Interdisciplinary Thematic Institute IMCBio, as part of the ITI 2021-2028 program of the University of Strasbourg, CNRS and Inserm, by IdEx Unistra (ANR-10-IDEX-0002), and by SFRI-STRAT'US project (ANR 20-SFRI-0012) and EUR IMCBio (ANR-17-EURE-0023) under the framework of the French Investments for the Future Program.

## Author contributions

**Kateryna Len-Tayon**: Conceptualization; Data curation; Formal analysis; Validation; Investigation; Visualization; Writing—original draft. **Justine Gantzer**: Data curation; Formal analysis; Investigation. **Charles Dariane**: Data curation; Investigation; Writing—original draft. **Olivier Fogel**: Data curation; Investigation; Writing—original draft. **Vanessa Friedrich**: Formal analysis; Writing—review and editing. **Daniela Rovito**: Conceptualization; Funding acquisition; Investigation; Writing—review and editing. **Véronique Lindner**: Formal analysis. **Valentine Gilbart**: Formal analysis. **Darya Yanushko**: Data curation; Formal analysis; Investigation. **Sandrine Henri**: Funding acquisition; Methodology. **Daniel Metzger**: Funding acquisition; Writing—original draft; Writing—review and editing. **Gilles Laverny**: Conceptualization; Data curation; Formal analysis; Supervision; Funding acquisition; Validation; Investigation; Visualization; Methodology; Writing—original draft; Writing—review and editing.

Source data underlying figure panels in this paper may have individual authorship assigned. Where available, figure panel/source data authorship is listed in the following database record: biostudies:S-SCDT-10_1038-S44321-026-00417-5.

## Disclosure and competing interests statement

The authors declare no competing interests.

# Expanded View Figures

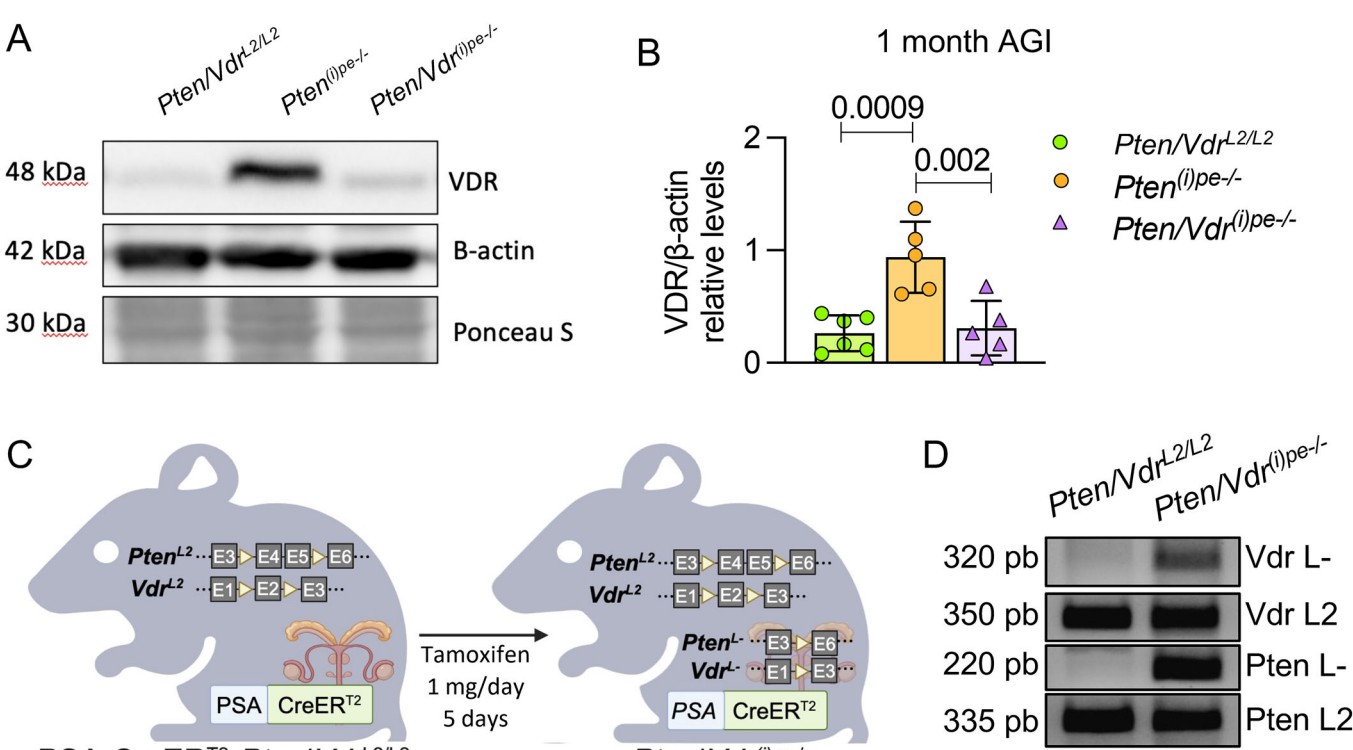

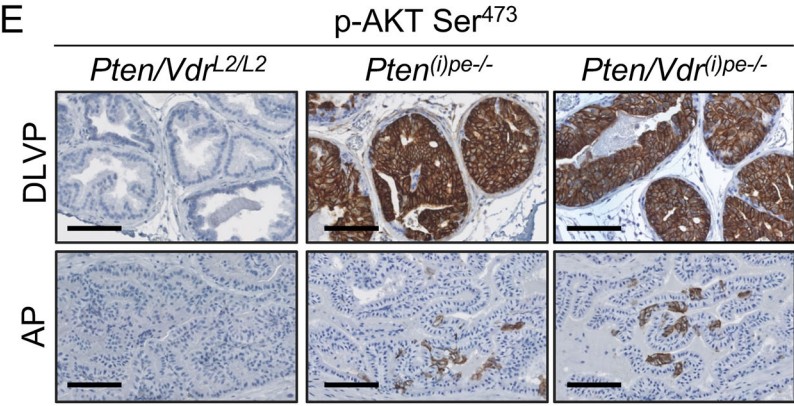

**Figure EV1. Generation and characterization of *Pten/Vdr*$^{(i)pe-/-}$ mice.**

Representative VDR immunoblot (**A**) and relative expression (**B**) determined in prostatic protein extracts from *Pten/Vdr*$^{L2/L2}$, *Pten*$^{(i)pe-/-}$ and *Pten/Vdr*$^{(i)pe-/-}$ mice 1 month AGI. β-actin was used as a loading control. $n = 5$–6 mice/group. Data are represented as mean $+/-$ standard deviation. $p$ value determined by two-way ANOVA with Tukey's post-hoc is indicated. (**C**) Schematic representation of the generation of *Pten/Vdr*$^{(i)pe-/-}$ mice. Yellow triangle represents LoxP sites. pe, prostatic epithelium; i, inducible; PSA, prostatic specific antigen; L2, floxed allele; L-, excised allele. (**D**) Representative image of the PCR products of the *Vdr* L-, *Pten* L-, *Vdr* L2 and *Pten* L2 alleles in DNA extracts from prostates of *Pten/Vdr*$^{L2/L2}$ and *Pten/Vdr*$^{(i)pe-/-}$ mice 1 month AGI. (**E**) Representative p-Akt Ser$^{473}$ immunostaining on DLVP and AP sections from *Pten/Vdr*$^{L2/L2}$, *Pten*$^{(i)pe-/-}$ and *Pten/Vdr*$^{(i)pe-/-}$ mice, 1 month AGI. Scale bar = 250 μm. $n = 4$ mice/group.

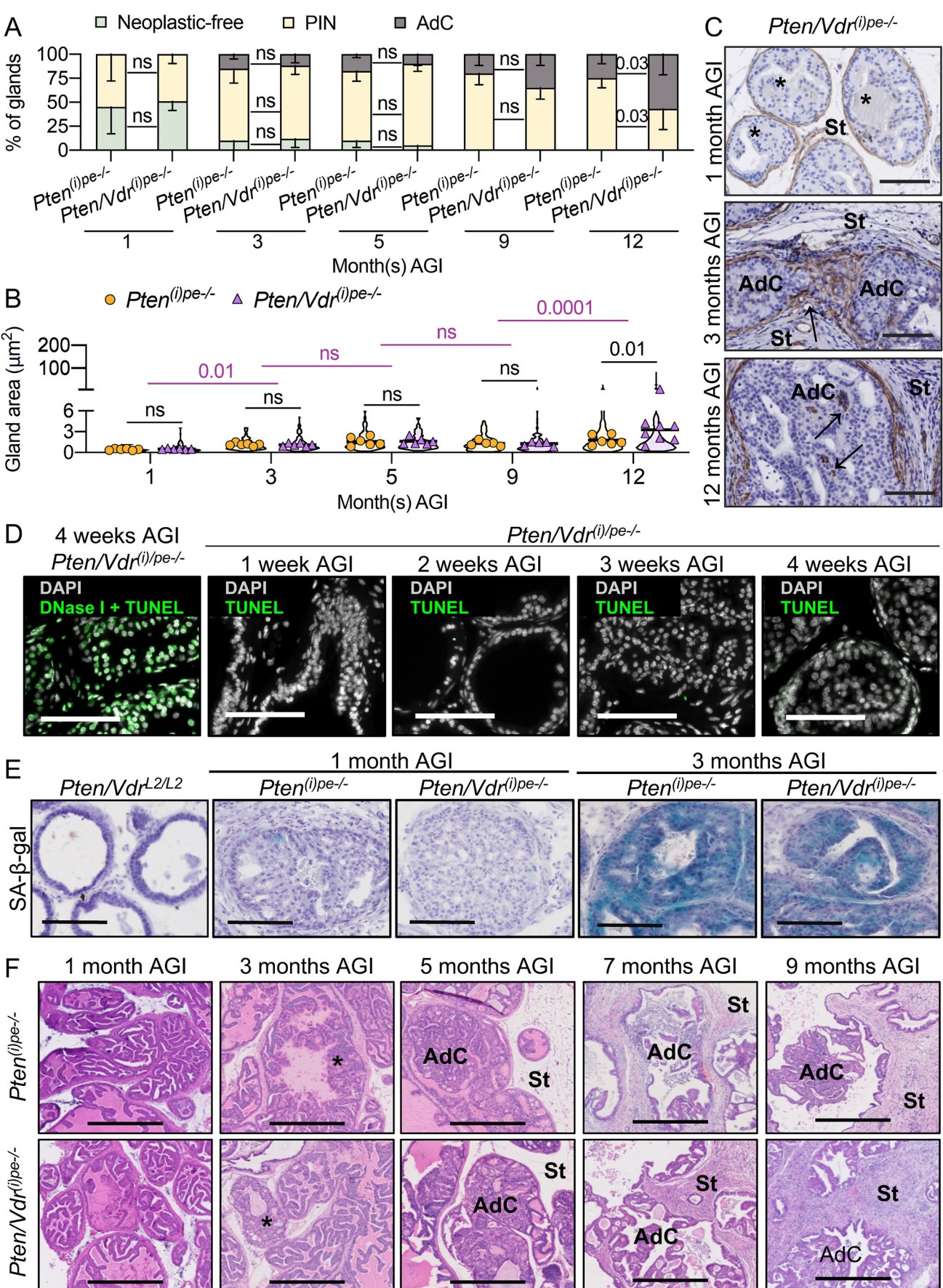

◀ **Figure EV2. Histological evaluation of prostatic tumors in *Pten^(i)pe-/-* and *Pten/Vdr^(i)pe-/-* mice.**

(A) Histological scoring of DLVPs from *Pten^(i)pe-/-* and *Pten/Vdr^(i)pe-/-* mice analyzed at 1, 3, 5, 9, and 12 months AGI. AdC, adenocarcinoma. $n = 5$ mice/group. Data are represented as mean - standard deviation. $p$ values determined by two-way ANOVA with Tukey's post-hoc are indicated. ns, $p \geq 0.05$. (B) Gland area quantification on DLVP sections from *Pten^(i)pe-/-* and *Pten/Vdr^(i)pe-/-* mice 1, 3, 5, 9, and 12 months AGI. $n = 7$ mice/group. $p$ values determined by two-way ANOVA with Tukey's post-hoc are indicated. ns, $p \geq 0.05$. (C) Representative α-SMA immunostaining on DLVP sections from *Pten/Vdr^(i)pe-/-* mice 1, 3, and 12 months AGI. Scale bar = 100 μm. $n = 3$ mice. *, PIN; →, invasion/adenocarcinoma; AdC, adenocarcinoma; St, stroma. (D) Representative images of TUNEL assay performed on DLVP sections from *Pten/Vdr^(i)pe-/-* mice 1-, 2-, 3-, and 4-week(s) AGI. DNAse I treated DLVP section was used a positive control. TUNEL/fluorescein (green) and DAPI (gray). Scale bar = 100 μm. $n = 2$ mice/group. (E) Representative SA-β-galactosidase staining of DLVP sections from *Pten/Vdr^L2/L2*, *Pten^(i)pe-/-* and *Pten/Vdr^(i)pe-/-* mice 1 and 3 months AGI. $n = 3$ mice/group. Scale bar = 250 μm. (F) Representative HE staining of AP sections from *Pten^(i)pe-/-* and *Pten/Vdr^(i)pe-/-* mice 1, 3, 5, 7, and 9 months AGI. Scale bar = 250 μm. $n = 5$ mice/group. *, PIN; AdC, adenocarcinoma; St, stroma.

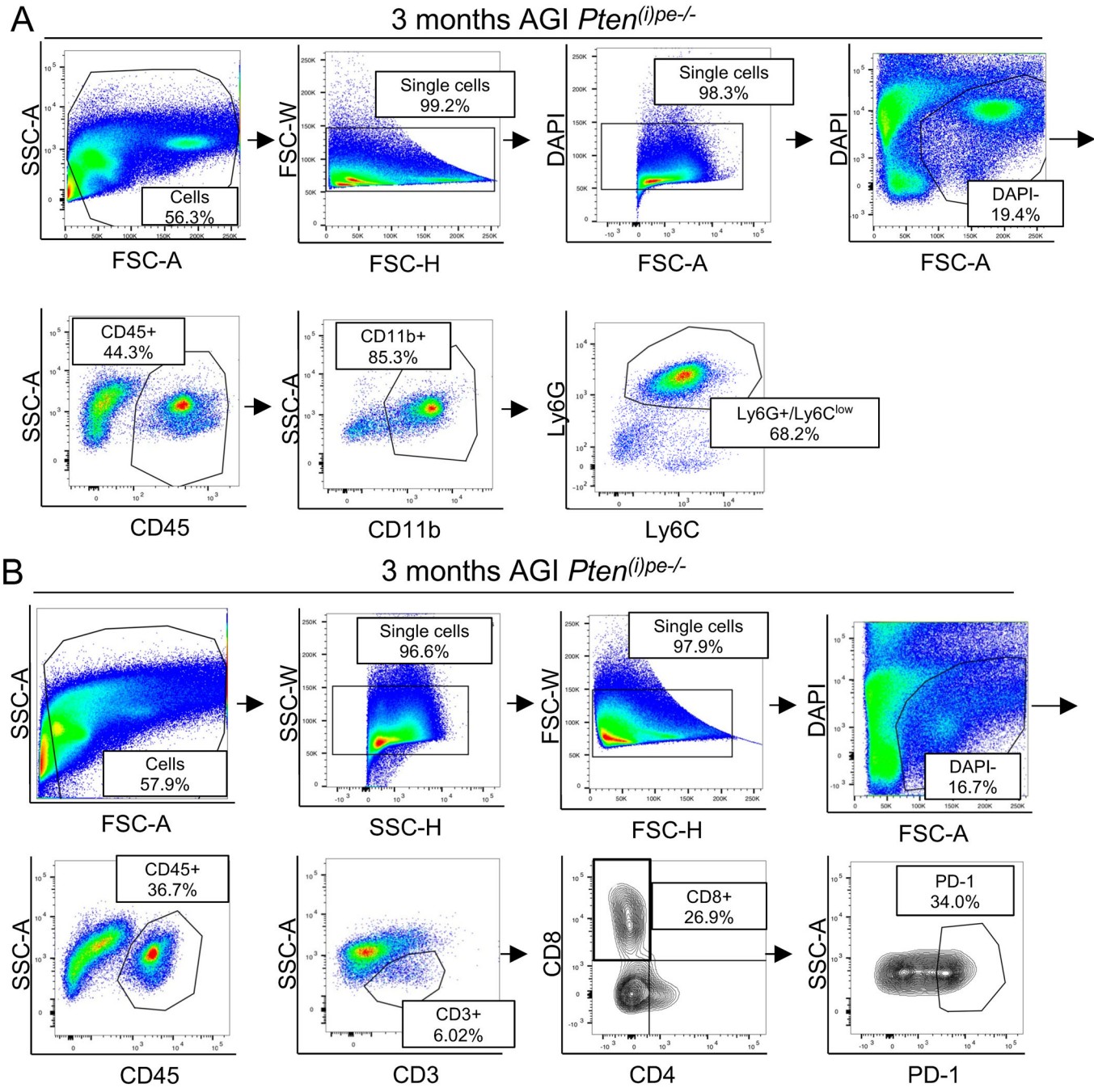

**Figure EV3.  Flow cytometry analysis of neutrophils and T cells.**

Representative gates for flow cytometry analysis of neutrophils (A) and T cells (B) in dissociated DLVPs from *Pten*(i)pe-/- mice 3 months AGI. *n* > 5.

## A

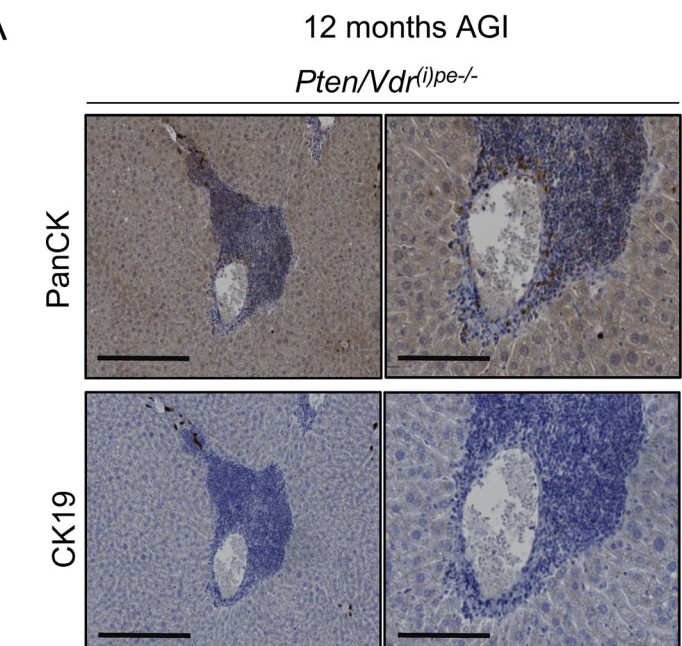

12 months AGI

*Pten/Vdr*<sup>(i)pe-/-</sup>

PanCK

CK19

## B

9 months AGI *Pten*<sup>(i)pe-/-</sup>

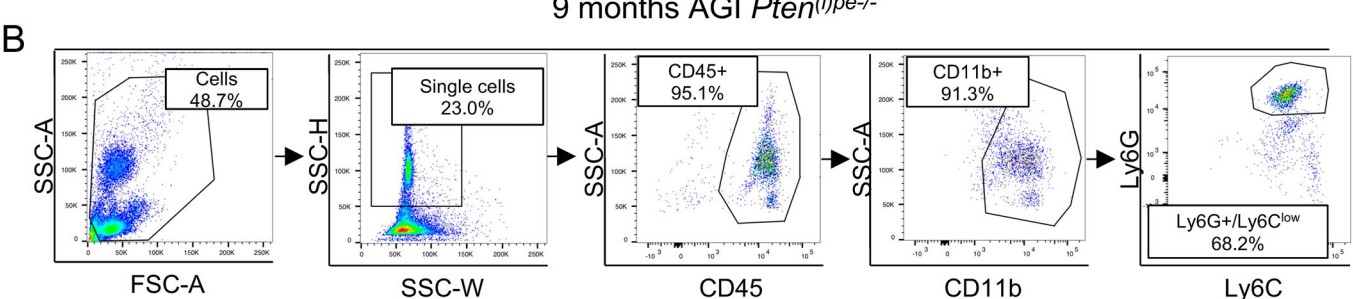

## C

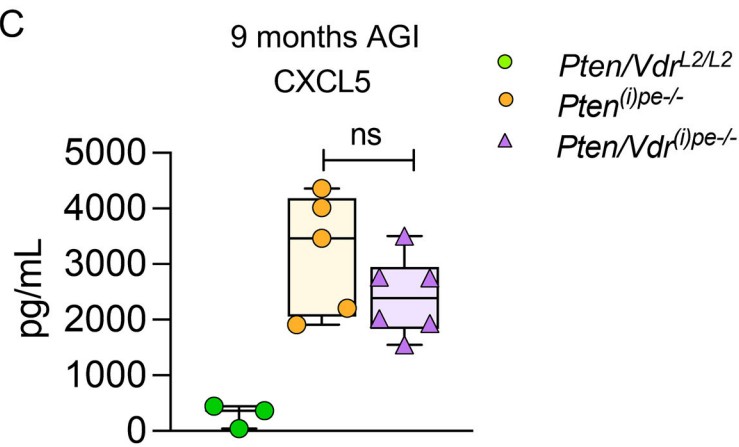

9 months AGI
CXCL5

ns

- 🟢 *Pten/Vdr*<sup>L2/L2</sup>
- 🟠 *Pten*<sup>(i)pe-/-</sup>
- 🔺 *Pten/Vdr*<sup>(i)pe-/-</sup>

**Figure EV4.   Characterization of liver metastasis and neutrophils infiltration.**

(**A**) Representative immunostaining of PanCK and CK19 on liver sections from *Pten/Vdr*<sup>(i)pe-/-</sup> mice 12 months AGI. Scale bar 250 μm (large view) and 50 μm (zoom). $n = 3$ mice. (**B**) Representative gates for flow cytometry analysis of neutrophils in blood from *Pten*<sup>(i)pe-/-</sup> mice 9 months AGI. $n = 8$. (**C**) CXCL5 levels determined in protein extracts from DLVPs of *Pten/Vdr*<sup>L2/L2</sup>, *Pten*<sup>(i)pe-/-</sup> and *Pten/Vdr*<sup>(i)pe-/-</sup> mice 9 months AGI. $n = 3$ *Pten/Vdr*<sup>L2/L2</sup>, $n = 5$ *Pten*<sup>(i)pe-/-</sup> and $n = 6$ *Pten/Vdr*<sup>(i)pe-/-</sup> mice. The boxes extend from the 25th to 75th percentiles, the lines represent the median, and the whiskers go down to the smallest up to the largest value. ns: not significant.

