## [Peer Review File · EMBO Molecular Medicine]

Impaired vitamin D signaling reveals neutrophils as key drivers of prostate cancer dissemination

Kateryna Len-Tayon, Justine Gantzer, Charles Dariane, Olivier Fogel, Daniela Rovito, Vanessa Friedrich, Veronique Lindner, Valentine Gilbert, Darya Yanushko, Sandrine Henri, Daniel Metzger, and Gilles Laverny

Corresponding author: Gilles Laverny (laverny@igbmc.fr)

Review Timeline:

Submission Date:	13th Apr 25
Editorial Decision:	14th May 25
Revision Received:	13th Jan 26
Editorial Decision:	18th Feb 26
Revision Received:	16th Mar 26
Accepted:	18th Mar 26

Editor: Zeljko Durdevic

Transaction Report:

14th May 2025

Dear Dr. Laverny,

Thank you for the submission of your manuscript to EMBO Molecular Medicine. We have now received feedback from the three reviewers who agreed to evaluate your manuscript. All three referees recognize interest of the study but also raise serious and partially overlapping concerns that should be addressed in a major revision. Considering that the revision will require extensive experimentation we think six months rather than three months would be more appropriate to provide the complete revision. If you would like to discuss further the points raised by the referees, I am available to do so via email or video. Let me know if you are interested in this option.

We would welcome the submission of a revised version within six months for further consideration. Please let us know if you require longer to complete the revision.

I look forward to receiving your revised manuscript.

Yours sincerely,

Zeljko Durdevic

Zeljko Durdevic
Senior Editor
EMBO Molecular Medicine

We require:

- 1) A .docx formatted version of the manuscript text (including legends for main figures, EV figures and tables). Please make sure that the changes are highlighted to be clearly visible.
- 2) Individual production quality figure files as .eps, .tif, .jpg (one file per figure). For guidance, download the 'Figure Guide PDF': (<https://www.embopress.org/page/journal/17574684/authorguide#figureformat>).
- 3) A .docx formatted letter INCLUDING the reviewers' reports and your detailed point-by-point responses to their comments. As part of the EMBO Press transparent editorial process, the point-by-point response is part of the Review Process File (RPF), which will be published alongside your paper.
- 4) A complete author checklist, which you can download from our author guidelines (<https://www.embopress.org/page/journal/17574684/authorguide#submissionofrevisions>). Please insert information in the checklist that is also reflected in the manuscript. The completed author checklist will also be part of the RPF.
- 5) Please note that all corresponding authors are required to supply an ORCID ID for their name upon submission of a revised manuscript.

6) It is mandatory to include a 'Data Availability' section after the Materials and Methods. Before submitting your revision, primary datasets produced in this study need to be deposited in an appropriate public database, and the accession numbers and database listed under 'Data Availability'. Please remember to provide a reviewer password if the datasets are not yet public (see <https://www.embopress.org/page/journal/17574684/authorguide#dataavailability>).

12) Author contributions: You will be asked to provide CRediT (Contributor Role Taxonomy) terms in the submission system. These replace a narrative author contribution section in the manuscript.

13) A Conflict of Interest statement should be provided in the main text.

14) Every published paper now includes a 'Synopsis' to further enhance discoverability. Synopses are displayed on the journal webpage and are freely accessible to all readers. They include a short stand first (maximum of 300 characters, including space) as well as 2-5 one-sentences bullet points that summarizes the paper. Please write the bullet points to summarize the key NEW findings. They should be designed to be complementary to the abstract - i.e. not repeat the same text. We encourage inclusion

of key acronyms and quantitative information (maximum of 30 words / bullet point). Please use the passive voice. Please attach these in a separate file or send them by email, we will incorporate them accordingly.

15) Include a Reagents and Tools Table as part of the Methods section, which can be downloaded from our author guidelines (<https://www.embopress.org/page/journal/17574684/authorguide#structuredmethods>)

**** Reviewer's comments ****

Referee #1 (Comments on Novelty/Model System for Author):

The work is generally well undertaken (the genomics is weaker) but in the end it's hard to tell if the work is impactful or not. Partly, this is because the authors don't really refer to the very large body of VDR prostate cancer work that has been undertaken (honestly, not mine! ...but it's written like this question hasn't been addressed) or the also large body of work using the Pten background to address the impact of other genes in prostate cancer. I wanted to know how these data reflect these other models. It could be the answer is Vdr is more modest, or it's really impactful. Either way is fine, but we should know after reading this.

Referee #1 (Remarks for Author):

1. Introduction - the VDR was cloned by Wes Pike originally, and should be cited (and not Bikle and Christakos review)
2. There has been a lot of work on VDR signaling in epithelial cells during prostate tumorigenesis (directly and indirectly, over the last 30 years). To say it has not been characterized is not accurate. The authors should choose some papers to cite
3. What is the genomic ancestry of the patients, especially given role of African ancestry to impact vitamin D function in the prostate
4. Did vitamin combine in a multivariate model to predict any clinical parameters?
5. F2 - for about 2500 DEGS it seems that the pvals for pathway enrichment were relatively modest - are these the most enriched terms? Likewise given there were up and down-regulated genes, were there no negative NES terms? Were there no cell cycle related terms? How were genes selected for labeling on volcano plot? Unclear how many biological replicates
6. Do the authors sense the 40% increase in proliferation is vdr -/- is explained by the impact of oxidation (given the impact of NAC is perhaps more modest?)
7. F3 clarify the regrouping to 9 clusters - is this from elbow plot or other analyses of variance type approaches? Please clarify then if this was a biased or unbiased approach to capture clusters? As written it sounds somewhat like cells were annotated with genes to form clusters? Was scRNA-Seq of Pten or Pten/vdr compared to a control, or between the two types? Unclear how many biological replicates
8. Did the authors consider pseudotime analyses in scRNA-Seq to illuminate impact of Vdr?
9. Unclear why FACS was undertaken to isolate luminal cells - why not analyze on DEGs from scRNA-Seq groups?
10. 120 genes intersecting from scRNA and bulk RNA-Seq is very small, given this is the same experiment essentially - please clarify any technical differences and statistical significance. Was there a larger overlap with luminal A? or A+C?
11. Given the small size of intersection (and subsetting by direction) it's probably not reasonable to undertake GSEA
12. Given that F3 F4 and F5 contains a lot of immune-orientated data then more of this should be in the introduction
13. In F4 can the authors should more precisely define "aggressive PCa"? It could be interpreted that the impact of Vdr is fairly modest.
14. Discussion - please compare this cross with Pten combinations such as p53, Smad4 or Gata3 - how does it compare?
15. The data don't appear to have been deposited and should be before publication.

Referee #2 (Remarks for Author):

In this manuscript, Len-Tayon and colleagues report an association between low peripheral blood levels of Vitamin D and PSA (a well-known prostate cancer marker, though not exclusively related to this pathology) as observed in a cohort of prostate cancer (PCa) patients upon initial diagnosis. Based on such observation, they proceed to model such a clinical condition relying on a conditional germline mouse model with luminal-specific Cre:ERT2-based excision of both Pten (a tumor suppressor frequently

lost in PCa patients) and Vdr (a key Vitamin D nuclear receptor) upon tamoxifen injection, at adult stages. The authors provide histological and molecular analyses over one year of age, including bulk and single-cell RNA-seq, flow-cytometry, and immunohistochemistry, with a particular focus on the tumor-associated recruitment of neutrophils at the primary tumor site, as well as in liver lesions. They test the ability of the CXCR1/CXCR2 inhibitor SX-682 to interfere with neutrophils recruitment, reprogram the tumor microenvironment, and slow down PCa progression. The topic is of considerable clinical interest, but several limitations complicate the interpretation of such findings as currently presented:

1. Clinical biomarker associations and statistics (related to Fig. 1)

- The association between blood level of Vitamin D and PSA in a cohort of patients with de novo PCa is not described with sufficient details, from both a statistical and a clinical point of view. The test applied to evaluate statistical significance is not reported, and the coefficient of determination appears to be modest. Regardless of statistical considerations, this analysis seems to refer to single timepoint measurements at the time of diagnosis. Vitamin D levels in the peripheral blood display significant seasonal and dietary oscillations on a monthly timescale, while PCa develops over many years, if not decades. The association between PSA levels and PCa is strong, but complex. Such statistical and clinical aspects challenge the interpretation provided by the authors, in the absence of further details on the biomarker acquisition and more in-depth data analysis.

2. Disease modeling (related to Figs. 1 and 2, EV Table 1)

- Regardless of point 1, a mouse model based on tamoxifen-inducible, PSA-CreER-T2-mediated excision of Pten and Vdr genomic alleles in luminal prostatic cells, doesn't model systemic low levels of Vitamin D. Vitamin D plays an important role in multiple endocrine and non-endocrine tissues, as well as in the immune system. Even restricting the focus on luminal prostatic cells, Vitamin D intracellular pathways are not exclusively transduced by Vdr, and non-Vdr-mediated pathways have been reported (e.g., Pdia3). Browsing the EV Table 1 related to bulk-RNAseq experiments performed by the authors on flow-sorted prostatic luminal cells in single conditional Pten^{-/-} vs. double conditional Pten^{-/-}; Vdr^{-/-} animals, the base median expression level of Vdr in control animals (Pten^{-/-}) is approximately 153 counts per million (CPM) reads (based on my guess, because I do not have access to the table legend) while for Pdia3 is 4270 (CPM). CPM are not normalized for gene length, nevertheless Pdia3 expression does not appear to be negligible. Additionally, the Vdr log₂FC in Pten^{-/-} vs. Pten^{-/-}; Vdr^{-/-} animals is slightly higher in the latter. This observation suggests the basal Vdr expression levels in Pten^{-/-} luminal prostatic cells are low, and thus the impact of genetic inactivation may be marginal. Functional experiments in organoids can help to clarify this issue.

3. Isolation of luminal prostate cells by flow cytometry (related to Figs. 2, 3, EV3)

- The flow cytometry strategy employed (Cd49f vs. Sca1) for isolating luminal prostate cells (especially, so called Luminal C cells displaying strong co-expression of both markers) do not clearly set apart basal from luminal cells. Adding an antibody for a luminal-specific surface antigen (e.g., Cd66a; PMID: 32915138) should improve the separation between these two cell identities. Based on a superficial evaluation of the tSNE representation (Fig. 3A), there seems to be a significant attrition of Luminal C cells in the final single-cell RNA-seq dataset. A table with absolute numbers/fraction of cell identities would clarify this point. A low number of Luminal C cells could make the follow-up analyses particularly noisy (Fig. 3B, C, D).

4. Characterization of cell identities (related to Fig. 3)

- The authors employ their own classification and nomenclature to characterize cell identities. At present, there is no widespread consensus in the field, and comparison with other datasets is recommended (e.g., PMID: 30566875, 32355025, 32915138).

5. Data presentation and statistics (related to Fig. 4)

- Figs. 1-3 of the manuscript focus on the analysis of the dorsolateral and ventral prostatic lobes (DLVP), perhaps because Akt activation is more prominent (Fig. EV1). In contrast to Fig. 1D, Fig. 4A also include the anterior lobe (AP) in the prostate mass quantification. However, some tumors arising in the AP may occlude the narrow prostatic duct causing the formation of very large cystic structures due to prostatic fluid accumulation more than tumor mass. In Fig. 4A, a few of such outliers may skew the analysis. Before applying the ANOVA, the authors should perform and report a test for homovariance across groups.

6. Detection of tumor infiltration in the liver (related to Fig. 4)

- The authors utilize a PanCK antibody for the detection of tumor infiltrates in the liver. However, this antibody isn't specific to tumor cells, and also mark cholangiocytes, especially upon ductular reaction in response to liver stressors or damage (e.g., PMID:31399003 (see Fig. 3)). A different strategy should be implemented to confirm the infiltration of PCa cells.

7. Data presentation and statistics (related to Fig. 5)

- A statistical test should be applied and reported for the drug study described in Fig. 5H, and Fig. 5I

8. Isolation of neutrophils by flow cytometry (related to Figs. 5, and EV4)

- It appears most single cells are lost during the flow cytometry experiment performed to isolate neutrophils (e.g., Fig EV4A and EV4B0). About 80-85% of single-cells are excluded upon gating for viable cells (DAPI-) suggesting potential issues during sample preparation. Such attrition could complicate the interpretation of follow-up analyses.

Referee #3 (Remarks for Author):

In this study, the authors provide evidence about the role of Vitamin D signaling in early prostate tumorigenesis and dissemination, correlating it with an aggressive phenotype. Using an evidence-based approach after obtaining clinical

information from a PCa patient cohort, they have conducted a comprehensive study using Pten/Vdr knockout in murine prostatic epithelial cells. Mechanistic insights suggest oxidative stress promotes tumor growth at an early stage in the PTEN-deficient mice model. Using single-cell and bulk RNA sequencing of the whole prostate and luminal cell population, respectively, the authors demonstrate the role of the CXCL5/CXCR2 axis in neutrophil chemotaxis, resulting in increased neutrophil dissemination and thereby, increase liver metastasis. While the study makes some contribution in identifying biomarkers for early detection of synchronous and metachronous visceral metastasis, the presented results do not adequately support the claims made. Following concerns are needed to be addressed:

Major Comments:

In Fig. 2, the authors have shown a positive association of Ki-67 proliferation marker with 8-OHdG. However, the mechanism is unclear, given that elevated 8-OHdG levels are also known to reduce cell proliferation (PMID: 22580124). An experiment demonstrating GC to TA transversion due to elevated 8-OHdG may be included to demonstrate the DNA damage induced proliferative changes.

The findings in Fig. 3N and Fig. EV4B demonstrate that an increase in exhausted T cell population in VDR-deficient mice is important for the conclusion of this study, and therefore, this needs to be supported by flow cytometry analysis in Pten/Vdr(i)pe^{-/-} mice.

While the authors demonstrate increased Cxcl5 expression in the Luminal C cell subpopulation, which may cause neutrophil chemotaxis, the results ultimately show no significant change in the infiltrating neutrophils in the prostate of 9-month AGI mice. It is unclear how circulating neutrophils are increased upon VDR-loss.

The authors should include a pathological score (eg, Gleason grade)-based comparative analysis of multiple patient cohorts to strengthen the rationale for the study.

The study should also include in vitro experiments (using cell lines, patient-derived organoids etc for overexpression and rescue experiments) to delineate the primary role of VDR signalling in PCa progression. Since these transgenic mice are the key model systems used throughout the study, it is advisable to include additional confirmatory experiments in other model systems for better accuracy.

The rationale behind single cell RNA sequencing of the dissociated prostate tissues is unclear in the study. Since the effect of impaired VDR signalling has not been delineated in sufficient detail, the authors should comment on the same to justify their hypothesis and expected outcomes.

While the authors highlight several key deregulated pathways because of VDR loss in PCa cells, they should experimentally conclude the effect of that dysregulation (eg, Sustained OS in PCa cells). How does sustained OS benefit PCa cells and aid their progression? The authors must validate such information using other experimental strategies or literature review-based studies.

"Moreover, flow cytometry analysis and immunostaining of (Ly-6G), the main neutrophil surface marker, revealed the absence of neutrophil infiltration at 1 month AGI, but their infiltration 3-months AGI was 2-fold higher in Pten/Vdr(i)pe^{-/-} DLVPs than in Pten(i)pe^{-/-}"

The authors should address the time lag in immunosuppressive neutrophil infiltration in PCa cells upon VDR loss. Concomitantly, the authors are urged to address the difference in biochemical staining (Ki-67, SASP, etc) between PTEN-null and PTEN/VDR-null cells at different time points (1,3, and 5 months).

Minor comments:

In Fig. 1, it is important to include the correlation between the treatment status of the patients along with other clinical parameters to clarify the relationship between Vitamin D levels and the disease in hormone-naive versus resistant patients. Appending the treatment information in the table might be helpful.

In Fig. EV1C, immunoblot demonstrating the levels of VDR in Pten/VdrL2/L2 control mice should also be included.

The authors are requested to chronologically align their figure calls in the manuscript and modify minor errors, if present. (Eg. Figure 3J legend)

Fig. 2G-J is incorrectly called as Fig. 3G-J in the manuscript.

There seems to be a typo error in the text demonstrating the role of SX-682 as a CXCR1/2 inhibitor.

The authors may summarise the mechanistic details concluded from this study in a schematic representation.

Some of the references are repetitive. For instance, Abu El Maaty MA, Grelet E, Keime C, Rerra AI, Gantzer J, Emprou C, Terzic J, Lutzinger R, Bornert JM, Laverny G, et al (2021) Single-cell analyses unravel cell type-specific responses to a vitamin D analog in prostatic precancerous lesions. *Sci Adv.* 7:eabg5982.

Referee #1 (Comments on Novelty/Model System for Author):

The work is generally well undertaken (the genomics is weaker) but in the end it's hard to tell if the work is impactful or not. Partly, this is because the authors don't really refer to the very large body of VDR prostate cancer work that has been undertaken (honestly, not mine! ...but it's written like this question hasn't been addressed) or the also large body of work using the Pten background to address the impact of other genes in prostate cancer. I wanted to know how these data reflect these other models. It could be the answer is Vdr is more modest, or it's really impactful. Either way is fine, but we should know after reading this.

We thank the reviewer for his/her comments and constructive suggestions.

Referee #1 (Remarks for Author):

1. Introduction - the VDR was cloned by Wes Pike originally, and should be cited (and not Bikle and Christakos review).

As suggested, we added the identification of the chicken VDR by Pr. Pike, as well as additional studies supporting that the effects of the bioactive vitamin D are mediated by VDR (page 4, line 82-84).

2. There has been a lot of work on VDR signaling in epithelial cells during prostate tumorigenesis (directly and indirectly, over the last 30 years). To say it has not been characterized is not accurate. The authors should choose some papers to cite.

As suggested, we added more details on previous works studying vitamin D signaling in PCa (page 4-5, lines 87-95).

3. What is the genomic ancestry of the patients, especially given role of African ancestry to impact vitamin D function in the prostate.

We agreed with the reviewer about the impact of African ancestry. Unfortunately, due to ethical and regulatory concerns, we had no access to the ethnic origin of the patients. We stated this limitation in the material and methods section (page 20, line 457).

4. Did vitamin combine in a multivariate model to predict any clinical parameters?

We apologized for the lack of precision regarding the analysis performed. A multivariate linear regression analysis using Graph Pad Prism V10 was used to determine the association among the various clinical parameters. This analysis showed that only 25(OH) vitamin D₃ and alkaline phosphatase (ALP) levels significantly correlate with the diagnostic marker PSA (Fig. 1A). The material and methods section (page 31, line 726) and the legend of Figure 1A (page 41, line 1022) have been amended accordingly.

5. F2 - for about 2500 DEGS it seems that the pvals for pathway enrichment were relatively modest - are these the most enriched terms? Likewise given there were up

and down-regulated genes, were there no negative NES terms? Were there no cell cycle related terms? How were genes selected for labeling on volcano plot? Unclear how many biological replicates

We thank the reviewer to pointing out the misleading information in Fig. 2C. The pathways presented belong to the top 10 of the up-regulated ones, ranked by the q-value, as shown in Table Extended View 2 of the original submission. The analysis was based on the q-value, corresponding to the p-value after the multiple hypothesis test correction. To enhance clarity, we amended Fig. 2C.

Even though some proliferation- and cell cycle-related genes were modulated (page 8, line 165), no cell cycle-related terms were enriched (see Table Extended View 2). As suggested, we have extended the description of the most negatively enriched terms (page 8, line 167-169).

The top deregulated genes enriched in the selected terms shown in Fig. 2C (Table Extended View 2) were depicted in the volcano plot. In addition, *Cdkn1a*, the p21-encoding gene, was added, as it represents a well-known VDR target in PECs.

The number of biological replicates (n = 3 mice/condition) has been clarified in the legend of Fig. 2A (page 41 line 1042).

6. Do the authors sense the 40% increase in proliferation is *vdr* ^{-/-} is explained by the impact of oxidation (given the impact of NAC is perhaps more modest?).

We agreed with the reviewer that the decrease in the proportion of 8OHdG positive cells might not be the only contribution to reduce the proliferation after NAC treatment. In this revised version, we confirmed that VDR-loss promotes mitochondrial respiration in PECs using VDR and/or PTEN-silencing in the non-malignant epithelial prostatic cells RWPE-1. As we show in a previous study that NAC reduces enhanced mitochondrial respiration-induced oxidative stress (PMID: 28623559), we propose that the effects of NAC is not limited to the reduction of 8OHdG positive cells, but it has also an impact on the mitochondria. We amended the manuscript accordingly (page 9 lines 187-200, Fig. 2K-N ; page 16 line 355)

7. F3 clarify the regrouping to 9 clusters - is this from elbow plot or other analyses of variance type approaches? Please clarify then if this was a biased or unbiased approach to capture clusters? As written it sounds somewhat like cells were annotated with genes to form clusters? Was scRNA-Seq of *Pten* or *Pten/vdr* compared to a control, or between the two types? Unclear how many biological replicates

Clusters were generated by an unbiased approach and then annotated based on prostatic-specific markers described in previous mouse datasets (PMID: 35867798 and 34330705). We improved the description of this section page 10,

line 223. In this study, a *Pten*^{(i)pe-/-} mouse was compared to a *Pten/Vdr*^{(i)pe-/-} mouse. The number of biological replicates (n = 1 mouse/condition) has been clarified in the figure legend 3 (page 42, line 1067).

8. Did the authors consider pseudotime analyses in scRNA-Seq to illuminate impact of Vdr?

The single cell analysis was performed at a single time point (3 months AGI), limiting the possibility to perform a pseudotime analysis.

9. Unclear why FACS was undertaken to isolate luminal cells - why not analyze on DEGs from scRNA-Seq groups?

scRNA-Seq is a potent technology to gain insights into changes in cellular heterogeneity. However, its limited sequencing depth constrains the identification of key regulatory pathways. Thus, scRNAseq was initially performed to assess differences in the tumor microenvironment between *Pten*^{(i)pe-/-} and *Pten/Vdr*^{(i)pe-/-} mice in an unbiased manner. Although DEGs in the luminal cell subset were initially analyzed using scRNA-seq data to identify candidate deregulated pathways (page 11, lines 235-238), this transcriptomic approach is known to only capture a subset of mRNA transcripts. Consequently, it did not allow for an in-depth analysis of the transcriptional differences between prostatic luminal cells of *Pten*^{(i)pe-/-} and *Pten/Vdr*^{(i)pe-/-} mice. Additionally, only one mouse was used for the scRNA-seq analysis. Therefore, to overcome these limitations, bulk RNA-seq analysis on FACS-isolated luminal cells was performed in biological triplicates and analyzed, providing higher transcript coverage and statistical robustness. As suggested, we provide an additional explanatory paragraph page 11, lines 240-243.

10. 120 genes intersecting from scRNA and bulk RNA-Seq is very small, given this is the same experiment essentially - please clarify any technical differences and statistical significance. Was there a larger overlap with luminal A? or A+C?

The modest overlap between the two datasets might result from differences in the sensitivity, sample preparation, and analytical workflows inherent to the two techniques (see the reply to point 9 above). While bulk RNA-seq generally provides a high sensitivity for gene expression, scRNA-seq typically allows for the analysis of a "transcriptomic signature", representing the most highly expressed genes within a given cell population. In addition, the 10X Genomics single-cell encapsulation platform is known to exhibit lower capture efficiency for luminal cells (PMID: 39865080), likely due to their senescence-related enlargement at this stage of cancer progression. Indeed, the proportion of luminal cells, particularly Luminal C, is 2-fold lower in the scRNA-seq dataset compared to flow cytometry analysis.

In addition, tissue sampling differences contributed to the observed variability: the entire prostate was processed for scRNA-seq, whereas only the DLVPs

were used for bulk RNA-seq. As shown in Appendix Fig. 2C, 95% of FACS-isolated cells from the DLVP are Luminal C cell. For this reason, the Luminal A cluster was not included in the comparison.

The DEGs from both scRNA-seq and bulk RNA-seq analyses were identified using a p-value < 0.05. In summary, the overlapping DEGs, which are enriched for the cellular response to oxidative stress and neutrophil recruitment, represent a robust core of biological changes consistently detected despite the technical and sampling differences.

Thus, the analyzed datasets are complementary. scRNA-seq details cell-specific changes across the whole prostate, while bulk RNA-seq provides an average expression profile from luminal cells in the DLVP.

11. Given the small size of intersection (and subsetting by direction) it's probably not reasonable to undertake GSEA.

We apologized for the misleading statements done in the manuscript. GSEA was performed on the bulk RNA-seq data, not on the intersection. We amended the manuscript page 11, line 250.

12. Given that F3 F4 and F5 contains a lot of immune-orientated data then more of this should be in the introduction.

As suggested, immunology-orientated information was added in the introduction page 4, lines 64-75.

13. In F4 can the authors should more precisely define "aggressive PCa"? It could be interpreted that the impact of Vdr is fairly modest.

As suggested, that was clarified page 13, lines 299-301 of the manuscript.

14. Discussion - please compare this cross with Pten combinattions such as p53, Smad4 or Gata3 - how does it compare?

As suggested, the phenotype of *Pten/Vdr*^{(i)pe-/-} mice were compared with that of the recently published *Pten/Tp53*^{(i)pe-/-} mice (page 18, lines 405-410), generated using a similar genetic strategy. Due to the different promoter used to induce gene inactivation, the comparison with other models, such as *Pten/Smad4*^{pc-/-} or *Pten/Gata3*^{pc-/-}, will potentially lead to confounding interpretations.

15. The data don't appear to have been deposited and should be before publication.

We amended the main text page 32, lines 729-732.

Referee #2 (Remarks for Author):

In this manuscript, Len-Tayon and colleagues report an association between low peripheral blood levels of Vitamin D and PSA (a well-known prostate cancer marker,

though not exclusively related to this pathology) as observed in a cohort of prostate cancer (PCa) patients upon initial diagnosis. Based on such observation, they proceed to model such a clinical condition relying on a conditional germline mouse model with luminal-specific Cre:ERT2-based excision of both Pten (a tumor suppressor frequently lost in PCa patients) and Vdr (a key Vitamin D nuclear receptor) upon tamoxifen injection, at adult stages. The authors provide histological and molecular analyses over one year of age, including bulk and single-cell RNA-seq, flow-cytometry, and immunohistochemistry, with a particular focus on the tumor-associated recruitment of neutrophils at the primary tumor site, as well as in liver lesions. They test the ability of the CXCR1/CXCR2 inhibitor SX-682 to interfere with neutrophils recruitment, reprogram the tumor microenvironment, and slow down PCa progression.

The topic is of considerable clinical interest, but several limitations complicate the interpretation of such findings as currently presented:

We thank the reviewer for his/her comments and constructive suggestions.

1. Clinical biomarker associations and statistics (related to Fig. 1)

- The association between blood level of Vitamin D and PSA in a cohort of patients with de novo PCa is not described with sufficient details, from both a statistical and a clinical point of view. The test applied to evaluate statistical significance is not reported, and the coefficient of determination appears to be modest. Regardless of statistical considerations, this analysis seems to refer to single timepoint measurements at the time of diagnosis. Vitamin D levels in the peripheral blood display significant seasonal and dietary oscillations on a monthly timescale, while PCa develops over many years, if not decades. The association between PSA levels and PCa is strong, but complex. Such statistical and clinical aspects challenge the interpretation provided by the authors, in the absence of further details on the biomarker acquisition and more in-depth data analysis.

As suggested by Reviewer (R)1 and R2, we improved the description of the statistic test used (page 31, line 726).

The coefficient of determination (R^2) is modest (0.1) for both positive control (ALP, Fig. 1B) and 25(OH) vitamin D₃ (Fig. 1C), likely reflecting high biological variability in PSA levels across patients, as well as the multifactorial nature of its regulation. Despite the modest R^2 , the observed association between PSA with ALP or vitamin D remains statistically significant. Given the clinical relevance of these correlations and the potential public health implications, it justifies further investigations.

As suggested, we have expanded the discussion about these results page 16, lines 365-378.

2. Disease modeling (related to Figs. 1 and 2, EV Table 1)

- Regardless of point 1, a mouse model based on tamoxifen-inducible, PSA-CreER-

T2-mediated excision of *Pten* and *Vdr* genomic alleles in luminal prostatic cells, doesn't model systemic low levels of Vitamin D. Vitamin D plays an important role in multiple endocrine and non-endocrine tissues, as well as in the immune system. Even restricting the focus on luminal prostatic cells, Vitamin D intracellular pathways are not exclusively transduced by *Vdr*, and non-*Vdr*-mediated pathways have been reported (e.g., *Pdia3*). Browsing the EV Table 1 related to bulk-RNAseq experiments performed by the authors on flow-sorted prostatic luminal cells in single conditional *Pten*^{-/-} vs. double conditional *Pten*^{-/-}; *Vdr*^{-/-} animals, the base median expression level of *Vdr* in control animals (*Pten*^{-/-}) is approximately 153 counts per million (CPM) reads (based on my guess, because I do not have access to the table legend) while for *Pdia3* is 4270 (CPM). CPM are not normalized for gene length, nevertheless *Pdia3* expression does not appear to be negligible. Additionally, the *Vdr* log₂FC in *Pten*^{-/-} vs. *Pten*^{-/-}; *Vdr*^{-/-} animals is slightly higher in the latter. This observation suggests the basal *Vdr* expression levels in *Pten*^{-/-} luminal prostatic cells are low, and thus the impact of genetic inactivation may be marginal. Functional experiments in organoids can help to clarify this issue.

Previous studies have investigated the systemic effects of vitamin D deficiency using dietary depletion models (PMID: 31028080), and we recently reported the impact of a vitamin D analog on PCa progression (PMID: 34330705). In contrast, the present study specifically aims to dissect the genomic effects of VDR signaling selectively in PECs, independently of its systemic roles in other tissues or in the immune system.

We acknowledge that basal *Vdr* transcript levels are lower than those of *Pdia3* in luminal cells, as observed in our bulk RNA-seq data. However, VDR protein levels are increased in PTEN-null PECs, as shown in Figure EV1A. The aim of this study was to unravel the role of VDR in prostatic epithelial cells during PCa evolution, and to further address it, we performed gene inactivation in non-malignant human prostatic epithelial cells RWPE-1. Our results showed that VDR-loss enhance the proliferation of PTEN-null RWPE-1 cells, consistent with the results seen in mice. These results highlight the importance of epithelial VDR in prostate cancer progression. However, we acknowledge in the discussion page 17 line 375-379 the emerging role of PDIA3 on vitamin D signaling. In addition, we amended the manuscript page 9, lines 186-201, add Figure 2K-N and Appendix Fig. 3B-C.

3. Isolation of luminal prostate cells by flow cytometry (related to Figs. 2, 3, EV3)

- The flow cytometry strategy employed (Cd49f vs. Sca1) for isolating luminal prostate cells (especially, so called Luminal C cells displaying strong co-expression of both markers) do not clearly set apart basal from luminal cells. Adding an antibody for a luminal-specific surface antigen (e.g., Cd66a; PMID: 32915138) should improve the separation between these two cell identities. Based on a superficial evaluation of the tSNE representation (Fig. 3A), there seems to be a significant attrition of Luminal C cells in the final single-cell RNA-seq dataset. A table with absolute numbers/fraction

of cell identities would clarify this point. A low number of Luminal C cells could make the follow-up analyses particularly noisy (Fig. 3B, C, D).

For the isolation of Luminal C cells, we employed a well-established gating strategy based on CD49f and SCA1. This approach remains a widely accepted standard for distinguishing basal and luminal cell populations in the prostate cancer research field (e.g., PMID: 28603917, 32355025).

As requested, we have added a table showing the absolute numbers and proportion of each cell type recovered in the single-cell RNA-seq dataset (Appendix Fig. 4). We agree with the reviewer that the proportion of luminal cells—particularly Luminal C—is lower in the scRNA-seq dataset compared to our FACS analysis. This discrepancy is consistent with previous reports, as the 10X Genomics single-cell encapsulation platform is known to exhibit lower capture efficiency for luminal cells (PMID: 39865080).

Importantly, this technical limitation did not compromise the objectives of the study, as the single-cell RNA-seq was primarily used to characterize general changes in the tumor microenvironment, rather than to analyze in-depth the molecular changes in epithelial subpopulations. To overcome this limitation and more specifically examine luminal epithelial cells, we complemented the single-cell data with bulk RNA-seq of FACS-isolated luminal cells.

4. Characterization of cell identities (related to Fig. 3)
• The authors employ their own classification and nomenclature to characterize cell identities. At present, there is no widespread consensus in the field, and comparison with other datasets is recommended (e.g., PMID: 30566875, 32355025, 32915138).

We adopted a classification system that we have previously detailed and validated (PMID: 35867798 and 34330705), which aligns with lineage markers commonly used in the field, including those described in the listed references (PMID: 30566875, 32355025, 32915138). Note that we chose to retain our nomenclature to emphasize key differences between our and other mouse models or human datasets, as well as to ensure consistency with our previous work.

5. Data presentation and statistics (related to Fig. 4)
• Figs. 1-3 of the manuscript focus on the analysis of the dorsolateral and ventral prostatic lobes (DLVP), perhaps because Akt activation is more prominent (Fig. EV1). In contrast to Fig. 1D, Fig. 4A also include the anterior lobe (AP) in the prostate mass quantification. However, some tumors arising in the AP may occlude the narrow prostatic duct causing the formation of very large cystic structures due to prostatic fluid accumulation more than tumor mass. In Fig. 4A, a few of such outliers may skew the analysis. Before applying the ANOVA, the authors should perform and report a test for homovariance across groups.

We acknowledge that prostatic fluid accumulation may greatly influence the tumor mass analysis. Note that, during the dissection process, most liquids were carefully removed. Nevertheless, we agree with the reviewer's comment and apologize for the use of ANOVA as the variances were heterogeneous. However, the histological grading confirmed that the severity of the tumors in *Pten/Vdr^{(l)pe/-}* mice is higher than in *Pten^{(l)pe/-}*. Thus, we removed the statistical analysis in Fig. 4A and amended the manuscript page 13, lines 290-293.

6. Detection of tumor infiltration in the liver (related to Fig. 4)
• The authors utilize a PanCK antibody for the detection of tumor infiltrates in the liver. However, this antibody isn't specific to tumor cells, and also mark cholangiocytes, especially upon ductular reaction in response to liver stressors or damage (e.g., PMID:31399003 (see Fig. 3)). A different strategy should be implemented to confirm the infiltration of PCa cells.

As suggested, we performed immunolabelling for cytokeratin 19 (CK19), a key cholangiocytes marker, on adjacent sections demonstrating that whereas the bile ducts are CK19 and PanCK positive, the disseminated cells are positive only for PanCK. These results have been implemented in the manuscript as Fig EV3A and in the text page 14, line 303.

7. Data presentation and statistics (related to Fig. 5)
• A statistical test should be applied and reported for the drug study described in Fig. 5H, and Fig. 5I

We acknowledge the absence of statistics in Fig. 5I. We now performed a Fisher's exact test to determine the significance of our findings. The Fig 5I was amended.

8. Isolation of neutrophils by flow cytometry (related to Figs. 5, and EV4)
• It appears most single cells are lost during the flow cytometry experiment performed to isolate neutrophils (e.g., Fig EV4A and EV4B0). About 80-85% of single-cells are excluded upon gating for viable cells (DAPI-) suggesting potential issues during sample preparation. Such attrition could complicate the interpretation of follow-up analyses.

Prostatic tissues are challenging to dissociate (an enzymatic + mechanic dissociation is classically applied in the field), resulting in the production of a significant proportion of debris during the process. Note that those debris, along with non-cellular aggregates, are excluded during the initial singlets and viability gating steps and do not impact the quality of the remaining viable single-cell population used for analysis. Neutrophils are a particularly sensitive cell type and are often among the first cells to undergo apoptosis or necrosis during tissue dissociation, if the procedure is not carefully optimized. The recovery of a high proportion of viable neutrophils (15-45% of DAPI- cells) in

our settings serves as an indicator of the quality and gentleness of our dissociation protocol.

Referee #3 (Remarks for Author):

In this study, the authors provide evidence about the role of Vitamin D signaling in early prostate tumorigenesis and dissemination, correlating it with an aggressive phenotype. Using an evidence-based approach after obtaining clinical information from a PCa patient cohort, they have conducted a comprehensive study using Pten/Vdr knockout in murine prostatic epithelial cells. Mechanistic insights suggest oxidative stress promotes tumor growth at an early stage in the PTEN-deficient mice model. Using single-cell and bulk RNA sequencing of the whole prostate and luminal cell population, respectively, the authors demonstrate the role of the CXCL5/CXCR2 axis in neutrophil chemotaxis, resulting in increased neutrophil dissemination and thereby, increase liver metastasis. While the study makes some contribution in identifying biomarkers for early detection of synchronous and metachronous visceral metastasis, the presented results do not adequately support the claims made. Following concerns are needed to be addressed:

We thank the reviewer for his/her comments and constructive suggestions.

Major Comments:

1. In Fig. 2, the authors have shown a positive association of Ki-67 proliferation marker with 8-OHdG. However, the mechanism is unclear, given that elevated 8-OHdG levels are also known to reduce cell proliferation (PMID: 22580124). An experiment demonstrating GC to TA transversion due to elevated 8-OHdG may be included to demonstrate the DNA damage induced proliferative changes.

To address the reviewer comments, we determined the GC to TA transversion in the RNA-seq data obtained from Luminal C cells 1 month AGI. The analysis showed that the percentage of transversion is similar in PECs from Pten^{(i)pe-/-} and Pten/Vdr^{(i)pe-/-} mice, suggesting that the proliferative changes are not induced by enhanced 8-OHdG-induced transversion. We amended the manuscript page 8, lines 177-179 and Appendix Fig. 3A.

2. The findings in Fig. 3N and Fig. EV4B demonstrate that an increase in exhausted T cell population in VDR-deficient mice is important for the conclusion of this study, and therefore, this needs to be supported by flow cytometry analysis in Pten/Vdr^{(i)pe-/-} mice.

We apologize for the confusion; the results depicted in Fig. 3N correspond to flow cytometry analysis. We amended the text page 12, line 275.

3. While the authors demonstrate increased Cxcl5 expression in the Luminal C cell subpopulation, which may cause neutrophil chemotaxis, the results ultimately show no significant change in the infiltrating neutrophils in the prostate of 9-month AGI mice. It is unclear how circulating neutrophils are increased upon VDR-loss.

We showed that VDR-loss drives an earlier and higher neutrophil infiltration at 3 months AGI, and proposed higher levels of the chemoattractant CXCL5 as underlying mechanisms. Indeed, at 9 months AGI, the neutrophils have been independently accumulated at the similar level in the DLVP of *Pten*^{(i)pe-/-} and *Pten/Vdr*^{(i)pe-/-} mice, and in line with that, we showed in this revised version that the levels of CXCL5 are similar in both mouse lines. These results sustain our previous report suggesting a direct association between CXCL5 levels and the extent of the neutrophil infiltrate. This suggests an effect in which the early VDR-loss promotes CXCL5 secretion driven a higher recruitment of neutrophils, and these levels together with the neutrophil infiltrate become similar at later time. These results are implemented in the manuscript as Fig EV4C and page 14, lines 324-325.

4. The authors should include a pathological score (eg, Gleason grade)-based comparative analysis of multiple patient cohorts to strengthen the rationale for the study.

As suggested a Gleason grade analysis was included as Appendix Fig 1 and page 6 lines 124-128.

5. The study should also include in vitro experiments (using cell lines, patient-derived organoids etc for overexpression and rescue experiments) to delineate the primary role of VDR signalling in PCa progression. Since these transgenic mice are the key model systems used throughout the study, it is advisable to include additional confirmatory experiments in other model systems for better accuracy.

To address the reviewer's comment, we silenced PTEN and/or VDR in the non-malignant human prostatic epithelial cell line RWPE-1 using siRNA-based approaches. As seen in mice, the inactivation of PTEN increased VDR levels and cell proliferation. Importantly, the proliferation of PTEN- and VDR-silenced cells was higher than that of cells with PTEN-silencing alone. Moreover, we showed that treatment with NAC has no effects on the proliferation of PTEN-silenced cells, but normalized that of PTEN- and VDR-silenced cells. These results, confirming those seen in mice, have been included into the manuscript as Fig. 2K-N and Appendix Fig. 3B-C and page 9 lines 187 – 200.

6. The rationale behind single cell RNA sequencing of the dissociated prostate tissues is unclear in the study. Since the effect of impaired VDR signalling has not

been delineated in sufficient detail, the authors should comment on the same to justify their hypothesis and expected outcomes.

Our primary hypothesis was that the loss of VDR signaling in PECs would induce cell type-specific changes within the prostate microenvironment and scRNA-seq analysis was essential to undercover those changes in an unbiased manner. Finally, we performed a complementary bulk RNA-seq to analyze in-depth the transcriptional differences in the luminal cell populations between *Pten*^{(i)pe^{-/-} and *Pten/Vdr*^{(i)pe^{-/-} mice. We amended the manuscript page 10, line 222.}}

7. While the authors highlight several key deregulated pathways because of VDR loss in PCa cells, they should experimentally conclude the effect of that dysregulation (eg, Sustained OS in PCa cells). How does sustained OS benefit PCa cells and aid their progression? The authors must validate such information using other experimental strategies or literature review-based studies.

Please refer to the reply of the comment 5 for the use of the NAC. We amended the discussion page 16 line 355.

8. "Moreover, flow cytometry analysis and immunostaining of (Ly-6G), the main neutrophil surface marker, revealed the absence of neutrophil infiltration at 1 month AGI, but their infiltration 3-months AGI was 2-fold higher in *Pten/Vdr*(i)pe^{-/-} DLVPs than in *Pten*(i)pe^{-/-}". The authors should address the time lag in immunosuppressive neutrophil infiltration in PCa cells upon VDR loss. Concomitantly, the authors are urged to address the difference in biochemical staining (Ki-67, SASP, etc) between PTEN-null and PTEN/VDR-null cells at different time points (1,3, and 5 months).

We wish to clarify that there is no time lag in the neutrophil recruitment between the *Pten*^{(i)pe^{-/-} and *Pten/Vdr*^{(i)pe^{-/-} mice. This population is indeed recruited at the same time point in both mouse lines, specifically between 2- and 3-months AGI. However, in *Pten/Vdr*^{(i)pe^{-/-} mice, neutrophil infiltration is approximately 2-fold higher 3 months AGI, which we attributed to the increased CXCL5 production by PECs, as shown by scRNA-seq, bulk RNA-seq, and ELISA (Fig 3 of the manuscript). We improved the description page 12 lines 265-270.}}}

Regarding biochemical staining, Ki67 levels were quantified at 1, 3, and 5 month(s) AGI and are presented in Fig. 1F-G. While other SASP factors besides CXCL5 were not individually determined, we performed a staining for SA-β-galactosidase, a well-established marker of senescence, which is shown in Fig. EV2E.

Minor comments:

9. In Fig. 1, it is important to include the correlation between the treatment status of the patients along with other clinical parameters to clarify the relationship between Vitamin D levels and the disease in hormone-naive versus resistant patients. Appending the treatment information in the table might be helpful.

The median time lag between the initiation of hormone therapy and the vitamin D blood test was 19 days, a time insufficient to observe therapy resistance. We amended the material and methods page 20 lines 450-452.

10. In Fig. EV1C, immunoblot demonstrating the levels of VDR in *Pten/Vdr*^{L2/L2} control mice should also be included.

We thank the reviewer for the suggestion to include VDR immunoblotting for *Pten/Vdr*^{L2/L2} control mice in Fig. EV1C. We did not directly compare *Pten/Vdr*^{L2/L2} with *Pten*^{(i)pe-/-} and *Pten/Vdr*^{(i)pe-/-} mice at 1 month AGI because, even at this early time point, we have already observed significant changes in the tumor microenvironment upon *Pten*-loss, including immune cell infiltration and fibroblast activation, which could contribute to the increased levels of VDR in total protein extracts from prostates (EV1C). However, as the *in vitro* data obtained within the framework of this revision showed that the loss of PTEN in non-malignant epithelial cells increases VDR protein levels, we implemented these results in Fig. EV1C-D and page 7 lines 138-141.

11. The authors are requested to chronologically align their figure calls in the manuscript and modify minor errors, if present. (Eg. Figure 3J legend)

We apologized for the inconvenience. We amended the manuscript to align the figures with the text.

12. Fig. 2G-J is incorrectly called as Fig. 3G-J in the manuscript.

As suggested, it was corrected page 9, line 186 of the manuscript.

13. There seems to be a typo error in the text demonstrating the role of SX-682 as a CXCR1/2 inhibitor.

As suggested, the text demonstrating the role of SX-682 as a CXCR1/CXCR2 inhibitor was corrected in page 15, line 331 of the manuscript.

14. The authors may summarise the mechanistic details concluded from this study in a schematic representation.

We added a graphical abstract to the revised version of the manuscript as requested.

15. Some of the references are repetitive. For instance, Abu El Maaty MA, Grelet E, Keime C, Rerra AI, Gantzer J, Emprou C, Terzic J, Lutting R, Bornert JM, Laverny G, et al (2021) Single-cell analyses unravel cell type-specific responses to a vitamin D analog in prostatic precancerous lesions. *Sci Adv.* 7:eabg5982.

The duplicated reference was removed.

18th Feb 2026

Dear Dr. Laverny,

Thank you for the submission of your revised manuscript to EMBO Molecular Medicine. I am pleased to inform you that we will be able to accept your manuscript pending the following final amendments:

1) In the main manuscript file, please do the following:

- Please address all comments suggested by our data editors listed below:

o Figure legends:

1. Please indicate the statistical test used for data analysis in the legends of figures 2A, B; 3D, G.

2. Please note that the box plots need to be defined in terms of minima, maxima, centre, bounds of box and whiskers, and percentile in the legends of figures 3H, J, N; 5A-C; J-L; EV4 C.

3. Please note that information related to n is missing in the legend of figure 3G.

4. Please note that the error bars are not defined in the legends of figures 2L, N; 4F, H; EV1 B, EV2 A.

5. Please note that the black arrows are not defined in the legend of figure 5P. This needs to be rectified.

- Add callouts for the Figure 3E.

- In Methods, provide the statement that informed consent was obtained from all human subjects and confirm that the experiments conformed to the principles set out in the WMA Declaration of Helsinki and the Department of Health and Human Services Belmont Report.

- Rename "Conflict of interest" to "Disclosure and competing interests statement". We updated our journal's competing interests policy in January 2022 and request authors to consider both actual and perceived competing interests. Please review the policy <https://www.embopress.org/competing-interests> and update your competing interests if necessary.

- Author contributions: Please remove it from the manuscript and specify author contributions in our submission system. CRediT has replaced the traditional author contributions section because it offers a systematic machine-readable author contributions format that allows for more effective research assessment. You are encouraged to use the free text boxes beneath each contributing author's name to add specific details on the author's contribution. More information is available in our guide to authors:

<https://www.embopress.org/page/journal/17574684/authorguide#authorshipguidelines>

- Remove the "List of abbreviations".

- In data availability statement please remove the sentence "Data are available from the Source Data file" and use the following format to report your deposited data and codes:

[data type]: [full name of the resource] [accession number/identifier] ([doi or URL or identifiers.org/DATABASE:ACCESSION])

Please check "Author Guidelines" for more information.

<https://www.embopress.org/page/journal/17574684/authorguide#availabilityofpublishedmaterial>

2) Source data: Please provide source data for Figure 3I. Currently, excel file Figure 1A-C is placed in Figure 3I folder.

3) Funding: Please merge it with Acknowledgement and make sure that information about all sources of funding are complete in both our submission system and in the manuscript. Ministry of Higher Education and Research, the Fondation pour la Recherche Medicale, INSERM/Pfizer Fellowship, ANR-10-752 IDEX-0002-02, ITI 2021-2028 program of the University of Strasbourg, CNRS and Inserm, ANR-10-IDEX-0002, ANR 20-SFRI-0012, ANR-17-EURE-002 are missing in our submission system.

4) Tables: Rename Tables EV1 - EV6 to Dataset EV1 - EV6, and Tables EV7 and EV8 to Table EV1 and EV2 and update their callouts in the main manuscript text. Please add a legend to current Tables EV7.

5) Appendix: Please add page numbers to the table of contents, correct the nomenclature to Appendix Figure S1 etc. and update the callouts in the main text.

6) The Paper Explained: Please provide "The Paper Explained" and add it to the main manuscript text. Please check "Author Guidelines" for more information. <https://www.embopress.org/page/journal/17574684/authorguide#researcharticleguide>

7) Synopsis:

- Synopsis image: Please resize the image 550 px-wide x 300-600 pixels high and upload it as a separate, high-resolution .jpeg file.

- Synopsis text: I have gone through your text and revised it (see below). Please review it and amend as you see fit:

Metastasis remains the leading cause of mortality in prostate cancer (PCa) patients, and the incidence of de novo hormone-naïve prostate cancer (mHNPC) is increasing. Our study provides mechanistic insights into how vitamin D signaling limits PCa progression and identifies promising therapeutic targets to prevent metastatic spreading in mHNPC.

Bullet points:

- Vitamin D receptor (VDR) signaling in prostatic epithelial cells (PECs) has protective roles and limits tumor progression.
- CXCL5 secretion by PTEN and VDR-deficient PECs promotes neutrophil infiltration.

- Circulating neutrophils are associated with tumor dissemination.
- Targeting neutrophil chemotaxis eliminates liver micro-metastases.

8) As part of the EMBO Publications transparent editorial process (see our Editorial at <http://embomolmed.embopress.org/content/2/9/329>), EMBO Molecular Medicine will publish online a Review Process File (RPF) to accompany accepted manuscripts. This file will be published in conjunction with your paper and will include the anonymous referee reports, your point-by-point response and all pertinent correspondence relating to the manuscript. Let us know if you want to remove or not any figures from it prior to publication. Please note that the Authors checklist will be published at the end of the RPF.

9) Please provide a point-by-point letter INCLUDING my comments as well as the reviewer's reports and your detailed responses (as Word file).

I look forward to reading a new revised version of your manuscript as soon as possible.

Yours sincerely,

Zeljko Durdevic

Zeljko Durdevic
Senior Editor
EMBO Molecular Medicine

*** Instructions to submit your revised manuscript ***

When preparing your revised manuscript, please refer to our guidelines: <https://link.springer.com/journal/44321/submission-guidelines#cms-Revised-submissions>. We perform an initial quality control of all revised manuscripts before re-review; failure to include requested items will delay the evaluation of your revision.

We require:

2) Individual production quality figure files as .eps, .tif, .jpg (one file per figure). For guidance, download the 'Figure Guide PDF': <https://media.springernature.com/original/springer-cms/rest/v1/content/27825798/data/v1>.

3) A .docx formatted letter INCLUDING the reviewers' reports and your detailed point-by-point responses to their comments. As part of the EMBO Press transparent editorial process, the point-by-point response is part of the Review Process File (RPF), which will be published alongside your paper.

4) A complete author checklist, which you can download from our author guidelines. Please insert information in the checklist that is also reflected in the manuscript. The completed author checklist will also be part of the RPF.

6) It is mandatory to include a 'Data Availability' section after the Materials and Methods. Before submitting your revision, primary datasets produced in this study need to be deposited in an appropriate public database, and the accession numbers and database listed under 'Data Availability'. Please remember to provide a reviewer password if the datasets are not yet public.

7) For data quantification: please specify the name of the statistical test used to generate error bars and P values, the number (n) of independent experiments (specify technical or biological replicates) underlying each data point and the test used to calculate p-values in each figure legend. The figure legends should contain a basic description of n, P and the test applied. Graphs must include a description of the bars and the error bars (s.d., s.e.m.).

9) Our journal encourages inclusion of *data citations in the reference list* to directly cite datasets that were re-used and obtained from public databases. Data citations in the article text are distinct from normal bibliographical citations and should directly link to the database records from which the data can be accessed. In the main text, data citations are formatted as follows: "Data ref: Smith et al, 2001" or "Data ref: NCBI Sequence Read Archive PRJNA342805, 2017". In the Reference list, data citations must be labeled with "[DATASET]". A data reference must provide the database name, accession number/identifiers and a resolvable link to the landing page from which the data can be accessed at the end of the reference.

12) Author contributions: You will be asked to provide CRediT (Contributor Role Taxonomy) terms in the submission system. These replace a narrative author contribution section in the manuscript.

13) A Conflict of Interest statement should be provided in the main text.

14) Every published paper includes a 'Synopsis' to further enhance discoverability. Synopses are displayed on the journal webpage and are freely accessible to all readers. They include a short stand first (maximum of 300 characters, including space) as well as 2-5 one-sentences bullet points that summarizes the paper. Please write the bullet points to summarize the key NEW findings. They should be designed to be complementary to the abstract - i.e. not repeat the same text. We encourage inclusion of key acronyms and quantitative information (maximum of 30 words / bullet point). Please use the passive voice. Please attach these in a separate file or send them by email, we will incorporate them accordingly.

15) Include a Reagents and Tools Table as part of the Methods section, which can be downloaded from our author guidelines.

Photos 400-800 DPI

*Additional important information regarding figures and illustrations can be found at
<https://media.springernature.com/original/springer-cms/rest/v1/content/27825798/data/v1>

***** Reviewer's comments *****

Referee #1 (Comments on Novelty/Model System for Author):

Given how druggable the VDR is, and its potential application in prostate cancer, then any study to identify mechanisms of sensitivity/efficacy and potentially impactful.

Referee #1 (Remarks for Author):

Collectively, it seems the three sets of reviews were complementary and comprehensive and the authors did a very good job of addressing them all.

Referee #2 (Remarks for Author):

The authors addressed most of the points raised during the first-round of revision and appropriately edited the discussion. Together with the reviewers' comments, this manuscript provides novel and useful insights into VDR signaling in prostate cancer.

The authors addressed the remaining editorial issues.

18th Mar 2026

Dear Dr. Laverny,

We are pleased to inform you that your manuscript is accepted for publication and is now being sent to our publisher to be included in the next available issue of EMBO Molecular Medicine.

You may qualify for financial assistance for your publication charges - either via a Springer Nature fully open access agreement or an EMBO initiative. Check your eligibility: <https://link.springer.com/journal/44321/how-to-publish-with-us>

Zeljko Durdevic
Senior Editor
EMBO Molecular Medicine

>>> Please note that it is EMBO Molecular Medicine policy for the transcript of the editorial process (containing referee reports and your response letter) to be published as an online supplement to each paper. If you do NOT want this, you will need to inform the Editorial Office via email immediately. More information is available here: <https://link.springer.com/partners/embo-press/editorial-policies#Peer%20review>